# Techno-Economic Analysis of a Seasonal Thermal Energy Storage System with 3-Dimensional Horizontally Directed Boreholes

Robert Beaufait, Willy Villasmil *, Sebastian Ammann and Ludger Fischer

Competence Center Thermal Energy Storage (CCTES), Lucerne University of Applied Sciences and Arts, Technikumstrasse 21, 6048 Horw, Switzerland
* Correspondence: willy.villasmil@hslu.ch

**Abstract:** Geothermal energy storage provides opportunities to store renewable energy underground during summer for utilization in winter. Vertically oriented systems have been the standard when employing boreholes as the means to charge and discharge the underground. Horizontally oriented borehole storage systems provide an application range with specific advantages over vertically oriented systems. They are not limited to the surface requirements needed for installation with vertical systems and have the potential to limit storage losses. Horizontal systems can be incorporated into the built environment and utilize underground storage sites below existing infrastructure. An experimental study examines configurations using a mix of renewable energy (photovoltaic panels) and grid energy to charge a storage system during summer for use during winter. A comparison of five different borehole configurations at three different loading temperatures was composed using an experimentally validated numerical model. The horizontal systems studied and analyzed in this work showed improved performance with scale and charging temperature. This paper supports further exploration into specific use cases for horizontal borehole thermal energy storage systems and suggests applications which would take advantage of better performance at scale.

**Keywords:** horizontal borehole; sensible thermal energy storage; seasonal storage; thermal grids

## 1. Introduction

Geothermal energy storage is a mature concept to store sensible heat seasonally which supports the decarbonization of building heat and domestic hot water. Current use of sensible seasonal thermal energy from underground takes various forms. This paper outlines the current use of the near-surface underground as a thermal energy storage medium and focuses on the use of three-dimensional, horizontally directed drilling (3D-HDD) for installation of a borehole thermal energy storage (BTES) system. Underground thermal energy storage is neither a new nor recently developed concept in the discipline of sensible seasonal thermal energy storage. Closed systems such as borehole heat exchangers, heat pipes, horizontal collectors, geothermal energy baskets and thermal piles have been used to extract heat from the ground [1]. Vertically oriented borehole heat exchangers have been used to extract low temperature heat from shallow depths and high temperature from deep depths (>3 km). Horizontal collectors, geothermal energy baskets and thermal piles extract low temperature heat in shallow depths from 1 to 10 m. A BTES extends the concept of a collector by charging the underground with heat for extraction at a later period. Storage of heat in the ground during periods of excess heat or energy (summer) for use during heat deficits (winter) is the scheme. Vertically oriented boreholes have also been employed to such ends [2–11]. However, they require direct access from above the storage site, thus limiting storage sites to those that are free of built environments or ecologically sensitive areas. A 3D-HDD BTES system can access storage sites belowground where the surface is not accessible with vertical drilling. It can be utilized to install horizontal

underground conduits where surface and underground conditions are limited by existing infrastructure or steep terrain. Technical and geological limitations, climate conditions and legal or legislative limitations must be considered before implementation [1,12]. Storage system applications include direct use of cold or low temperature heat with and without a heat pump, a system incorporating solar collectors, combined heat and power plants and systems installed below sealed surfaces, which could be used as a source of heat in summer. Figure 1 below shows the temperature ranges for storing heat and cool belowground for different applications [13].

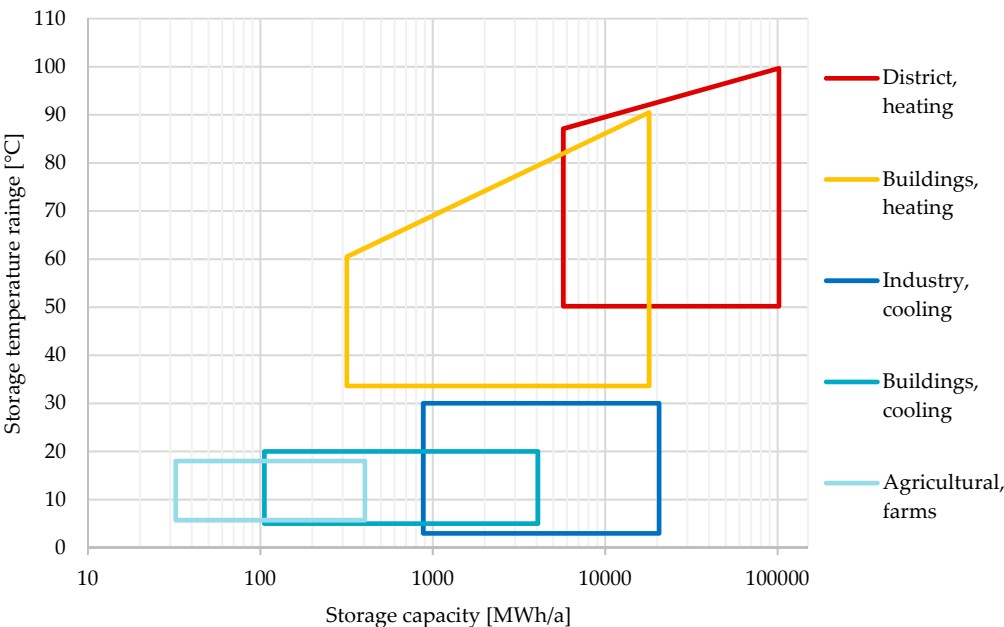

**Figure 1.** Storage capacity and temperature ranges for different sectors according to the VDI [13].

The temperature and capacity levels of the BTES systems considered in this study are in the range of around 40 °C and between 100 $MWh_{th}$/a and 4.5 $GWh_{th}$/a. A test site was constructed to conduct a thermal response test and validate the numerical model developed in this work. The experimentally validated model was expanded to allow a more comprehensive techno-economic analysis for larger configurations and different charging temperatures. Boreholes were spaced between 1.5 and 3 m in a pattern consistent with former vertical installations [14,15]. Five different configurations with 4, 7, 12, 24 and 42 boreholes were modeled with COMSOL Multiphysics [16]. The focus of this work is outlined by the following points.

- Decarbonization of building heat using renewable energy and seasonal thermal energy storage.
- The use of the underground below established surface structures as a storage medium.
- Determine the economic feasibility of a new application for storing heat underground.

## 2. Materials and Methods

An experimental site to charge, discharge, and record measurements was installed at Heldswil, Switzerland. The installation was used to determine soil properties and record the evolution of temperature in the surrounding soil from a buried heat source and heat sink. A portion of the measurement results were used as validation for a simplified 3D numerical model of what was installed and measured at the test site. The validated model was expanded to simulate multi-borehole configurations for thermal performance (Section 4.1) and economic assessment for a reference building stock supplied by a district heating network (DHN) (Section 4.2).

## 2.1. 3D-HDD Applied to BTES Systems

Five different borehole configurations using three charging temperatures were evaluated by charging the ground with heat in summer and discharging the heat in winter. Variability of the underground exists regarding material properties, spatial distribution of individual materials, and measurement methods to determine thermal properties of an inhomogeneous material [17]. This study used data for average material properties of the underground representative of the test site.

Underground anomalies, loose or disturbed earth, porous media, and flowing, stagnant, or seasonal groundwater features were not in the scope of this work. Average values for fixed moraine stated below in Table 1 were applied for the entire underground. In addition, the interface between the underground and the aboveground environment is important as it represents an important thermal boundary. This study assumed an idealized ground surface that experienced a change in temperature throughout the year reflective of the local air temperature.

**Table 1.** Soil and material properties [18].

| Rock/Soil Type | ($\lambda$) W/(m·K) | | ($C_{p,v}$) MJ/m$^3$ | | ($\rho$) 10$^3$ kg/m$^3$ |
|---|---|---|---|---|---|
| | **Recom** | **Calc** | **Recom** | **Calc** | |
| clay, dry | 0.4–1.0 | 0.6 | 1.5–1.6 | 1.5 | 1.8–2.0 |
| clay, saturated | 0.9–2.3 | 1.4 | 2.0–2.8 | 2.3 | 2.0–2.2 |
| sand, dry | 0.3–0.8 | 0.5 | 1.3–1.6 | 1.4 | 1.8–2.2 |
| sand, saturated | 1.5–4.0 | 2.3 | 2.2–2.8 | 2.4 | 1.9–2.3 |
| gravel/stone, dry | 0.4–0.5 | 0.4 | 1.3–1.6 | 1.4 | 1.8–2.2 |
| gravel/stone, saturated | 1.6–2.0 | 1.7 | 2.2–2.6 | 2.3 | 1.9–2.3 |
| fixed moraine | 1.7–2.4 | 1.8 | 1.5–2.5 | 2.0 | 1.9–2.5 |
| peat | 0.2–0.7 | 0.4 | 0.5–3.8 | 1.6 | 0.5–0.8 |

Equation (1) below is a relationship for the solution of the 1-D heat equation for underground soil with Table 2 defining the parameters [19].

$$T_{soil}(z,t_s) = T_m(z,t_s) - T_p(\text{lat/long}) \cdot e^{-z\sqrt{\frac{\omega}{2\alpha}}} \cdot \cos\left(\omega t_s - \varphi - z\sqrt{\frac{\omega}{2\alpha}}\right) \tag{1}$$

**Table 2.** Parameter description for heat equation in the underground [19].

| Parameter | Description | Unit |
|---|---|---|
| $T_{soil}(z,t_s)$ | Soil temperature as a function of depth $z$ and time $t_s$. | K |
| $T_m(z,t_s)$ | Initial temperature at a depth $z$ and time $t_s$. | K |
| $T_p(\text{lat/long})$ | Annual amplitude of the monthly average temperature cycle at a given location. | K |
| $z$ | Depth under the surface. | m |
| $\alpha$ | Thermal diffusivity of soil. | m$^2$/s |
| $\omega$ | Frequency of cycle (1 year). | rad/s |
| $t_s$ | Time elapsed from Jan 1st. | s |
| $\varphi$ | Phase shift from Jan 1st for coldest/hottest day of the year. | rad |

Figure 2 below shows an approximate belowground profile by plotting Equation (1) for a homogenous underground soil at the geographical location of the test site.

In the solar energy zone (0~10 m) and the geo-solar transition zone (10~40 m), a pronounced temperature shift can occur from the sealing of the surface to the environment (buildings, roads, parking lots, etc.) [1].

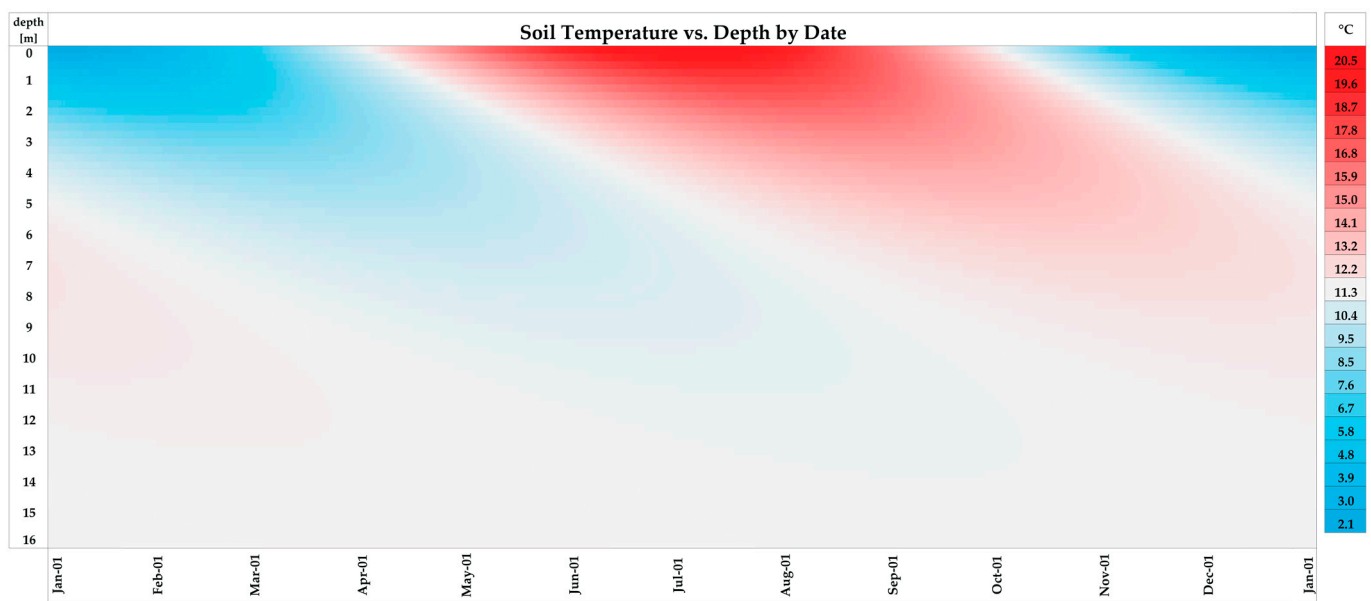

**Figure 2.** Generalized soil temperature with depth and date with no surface coverage at the test site.

### 2.2. Experimental Test Site

The test site was constructed to record the temperature of the soil and validate the numerical model developed in this work. A total of 11 boreholes were drilled under driveways, underneath and adjacent to aboveground structures (light gray), underneath and adjacent to a ramp (medium gray), and next to a warehouse with an insulated belowground floor (dark gray). Figure 3 below shows a 3D rendering of these features relative to the boreholes from two perspectives.

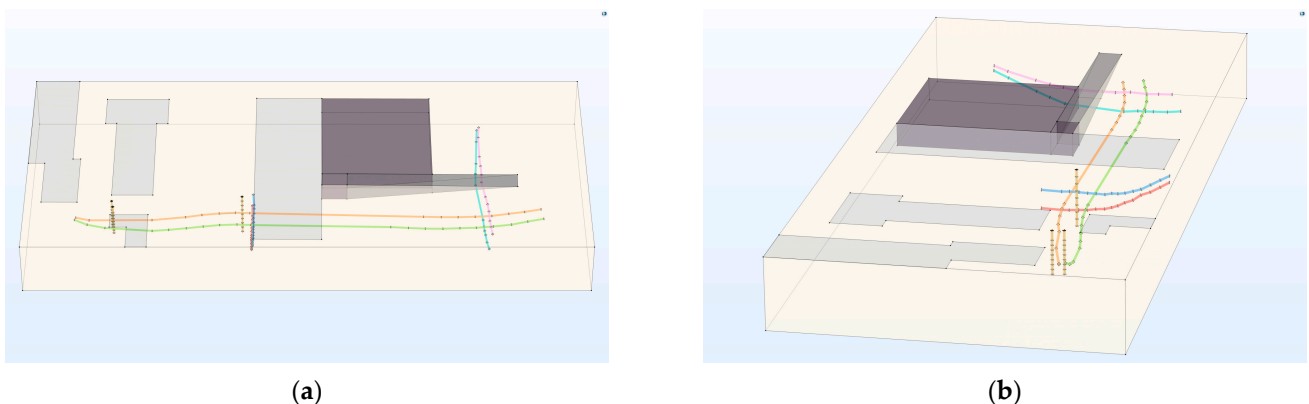

(**a**)                                                                                        (**b**)

**Figure 3.** Lengthwise (**a**) and diagonal (**b**) perspectives of the experimental boreholes.

Two parallel horizontal boreholes (BOHR1 and BOHR2) were drilled with a length of 126 m. Additionally, 4 horizontal-orthogonal (BOHR3–BOHR6) and 3 vertical-orthogonal (KERN1–KERN3) boreholes were drilled around the heated borehole (BOHR2) for temperature measurement with an additional 4 vertical boreholes for soil sampling. Figure 4 below shows an aerial perspective of the 126 m-long boreholes and the additional boreholes used for recording temperature overlaid on the property layout. Each borehole and sensor ID are labeled with a color-coded name corresponding to the borehole (e.g., BOHR3, 210, V1_20), and sensors are indicated with a small yellow-and-magenta diamond (⧫).

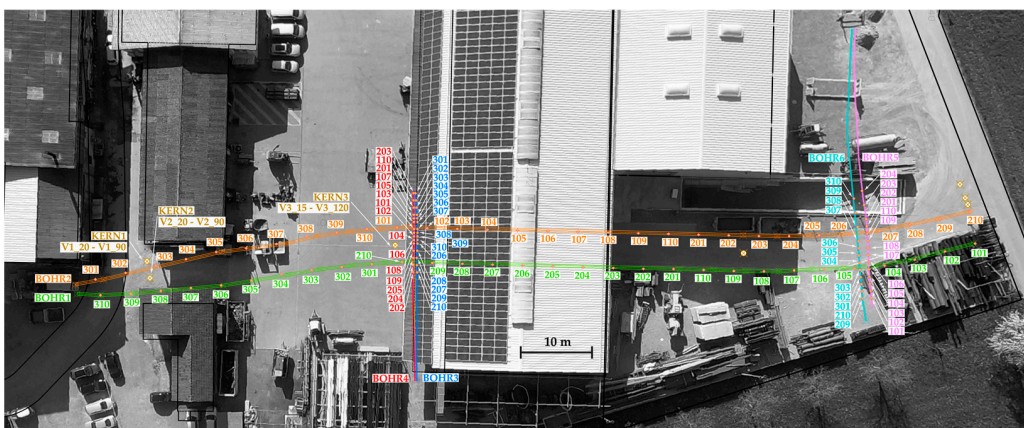

**Figure 4.** Layout of boreholes and temperature sensors installed on the property of Schenk AG.

The heated borehole (BOHR2) ran parallel to the lengthwise measurement borehole (BOHR1) which allowed temperature measurements at different distances from the entry point to the ground for the entire 126 m. Sensors in BOHR2 were rendered invalid due to imprecise position in a region of sharp temperature gradients. Orthogonal boreholes KERN1–KERN3 enabled vertically oriented temperature measurements at specified distances from the entry point to the ground. BOHR1 and BOHR2 required a sloped entry length whereby the drilled borehole was nearer to the surface than the target depth. Figure 5 below shows the underground profile of BOHR1, BOHR2, KERN1, KERN2, and KERN3 as they were drilled. Boreholes KERN1–KERN3 also include sensor IDs and location using the same convention as in Figure 4 above.

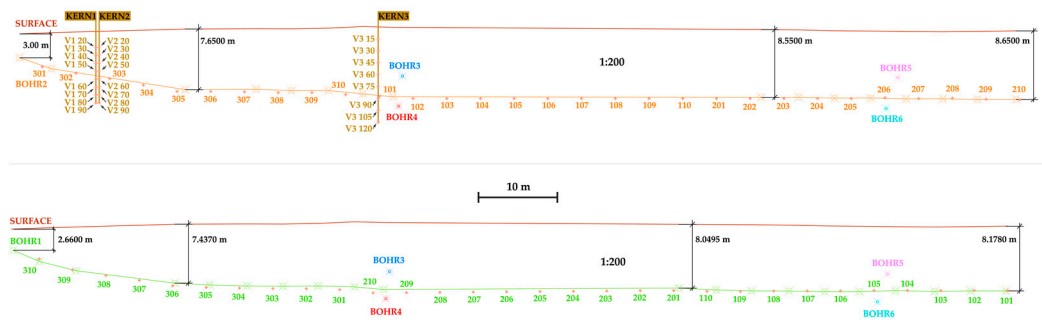

**Figure 5.** Profiles of BOHR1 and BOHR2 with vertical and horizontal temperature sensors.

Orthogonal boreholes BOHR3–BOHR6 enabled horizontally oriented temperature measurements above and below the heated borehole at specified distances from the entry point to the ground. Figure 6 below shows the distribution of temperature measurements sensors in BOHR3–BOHR6 relative to BOHR1 and BOHR2 from the perspective of the entry point.

BOHR1 and BOHR2 enclosed high-density polyethylene pipes used for different purposes. The heated borehole (BOHR2) contained two pipes: a coaxial pipe containing the heat transfer fluid (HTF) (red and blue), and a regular pipe used for measuring temperature (green). The parallel measurement borehole (BOHR1) was cased in a 200 mm-diameter pipe with a second pipe for housing the temperature sensors. All air space in both the borehole (BOHR2) and pipe (BOHR1) were filled with bentonite (gray). Pipe walls are shown in dark gray. Dimensions of the pipes are shown in Figure 7. The coaxial inner pipe (blue) and annulus (red) had wall thicknesses of 4 mm and 5.8 mm, respectively. Both measurement conduits (green) had wall thicknesses of 5.8 mm, while the 200 mm pipe in BOHR1 had a wall thickness of 18.2 mm. The spacing between BOHR1 and BOHR2 ranged between 2 and 5 m drilled to a depth of ~2.5 to ~8.5 m.

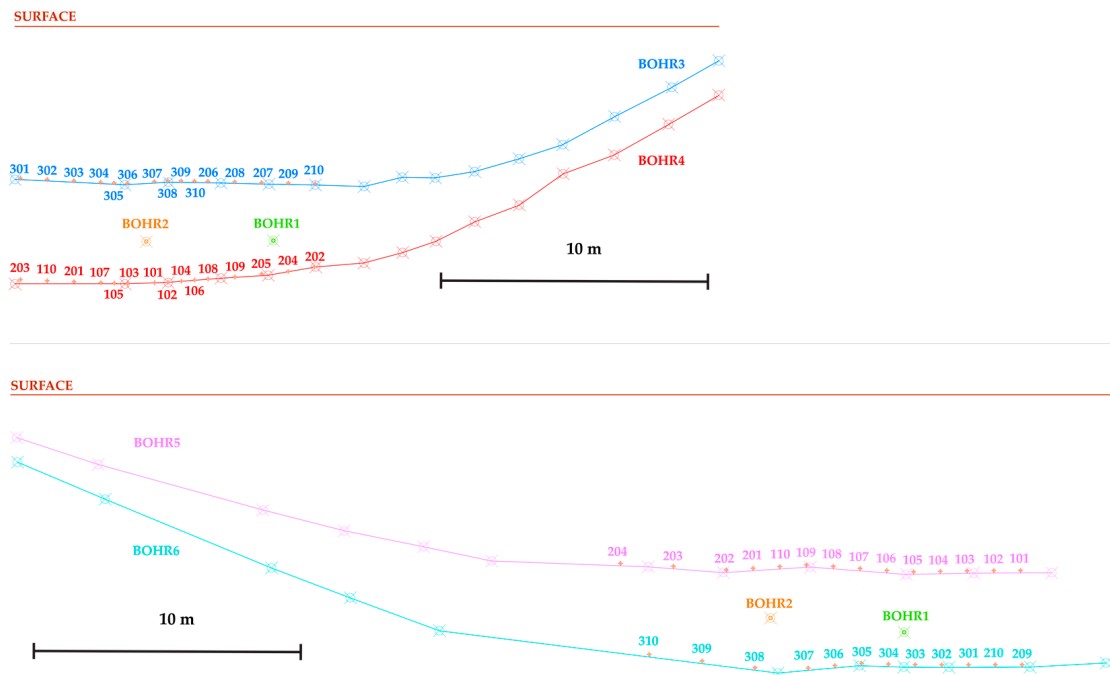

**Figure 6.** Profiles of BOHR3–BOHR6 with temperature sensors relative to BOHR1 and BOHR2.

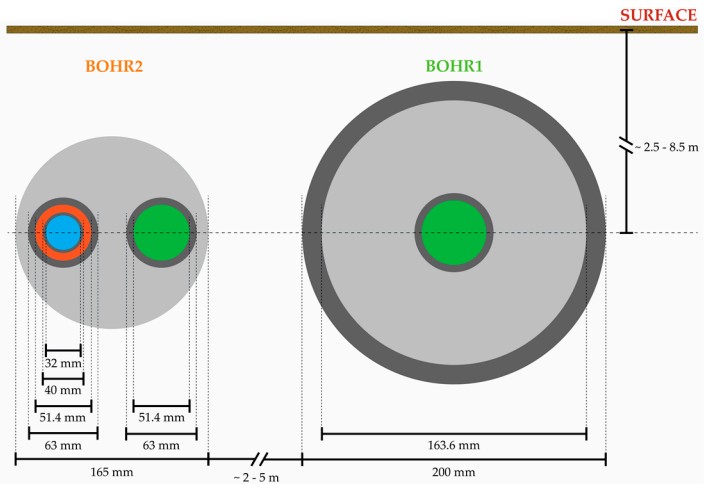

**Figure 7.** Cross–section of boreholes of BOHR1 and BOHR2 with internal piping.

BOHR1 and BOHR2 each had 30 PT100 DIN1/3 temperature sensors installed (green pipes) and spaced ~3.7 m to ~5 m apart. Orthogonal measurement boreholes (BOHR3–BOHR6) each had 15 PT100 DIN1/3 temperature sensors installed (blue, red, pink, cyan lines in Figures 4–6) and spaced ~0.5 to ~2 m apart.

Vertical boreholes were drilled to obtain soil samples to analyze properties of the soil, with three of them being used as measurement conduits (KERN1–KERN3) marked with a gold circle and crosshairs (⊠). Each of the three vertical boreholes used for measurement had 8 temperature sensors spaced ~1 and ~1.5 m apart. All temperature sensors were recorded with an Agilent 34410A/11A multimeter. The material properties of components used in the construction of the heated borehole and measurement network are listed in Table 3.

**Table 3.** Material properties of equipment used for the experimental set-up.

| Product | Material | $C_p$ kJ/(kg·K) | $\lambda$ W/(m·K) | $\rho$ kg/m³ |
|---|---|---|---|---|
| pipe | HDPE [20,21] | 1.8 | 0.4 | 960 |
| filling material | bentonite [22] | 1.2 | 1.2 | 2000 |
| soil | fixed moraine [18] | 0.9 | 2.0 | 2000 |
| HTF | water * [23] | 4.195–4.183 | 0.58–0.65 | 998–983 |

* Range of temperature dependent values between 10 and 60 °C.

Each phase of operation was recorded for the inlet temperature, outlet temperature, and volume flow of the heated borehole (BOHR2). Table 4 below lists the names, types, and duration for the period of experimental operation (see the first figure, Section 3.1.).

**Table 4.** Name, type, and duration of operating periods of the experimental set-up.

| Name | Type | Start | End |
|---|---|---|---|
| H1 | heating | 10 May 2019 | 26 July 2019 |
| D1 | drift | 27 July 2019 | 14 October 2019 |
| H2 | heating | 15 October 2019 | 24 February 2020 |
| D2/R1 | drift/recirculation | 25 February 2020 | 1 June 2020 |
| H3 | heating | 2 June 2020 | 18 September 2020 |

The HTF was pumped with a mass flow between ~0.1 and ~0.3 kg/s and an inlet temperature of 60 °C.

### 2.3. Modeling a BTES System

The development of a numerical model was necessary to evaluate the performance of the borehole thermal energy storage system during different phases of operation. Diffusive thermal effects within the storage medium and a transient boundary condition required numerical techniques to assess both the thermal behavior of heat flow from an underground source to the surrounding earth during charging and heat flow to an underground sink during discharging. A simplified approach to estimate the performance of the experimental borehole was dimensioned with a straight 127 m-long coaxial pipe representative of the heated pipe in BOHR2 through a homogenous block of fixed moraine measuring 30 × 137 × 20 m. The simplified model did not account for anomalous surface and belowground structures, the entry length needed to attain the target depth, nor the influence of water. An average depth of 7.65 m for the experimental borehole was calculated and applied to the model. The numerical model was implemented using the finite element modeling software COMSOL 6.0 using heat transfer in solids and fluids with 185,334 elements [16].

Adiabatic boundary conditions were set on all vertical walls of the domain with an isothermal boundary of 11.3 °C set on the floor of the domain. The top of the domain (earth surface) was a transient boundary that was set to the local average daily temperature. The initial conditions of the earth were governed according to the local climate and the relationship found in Equation (1) (Section 2.1). Figure 8 above shows a diagram of the boundary and initial conditions with dimensions and flow pattern.

15 April was the start date of the 5-month charging period for the BTES system in all modeled configurations except for the single-bore validation model. Heat produced with a heating cell was transferred during an 8-h period between 11:00 and 19:00. These hours were chosen for having the highest temperatures during the day with significant sunlight. Weather parameters used in this study were sourced from Meteonorm [24]. Figure 9 shows the distribution of daily temperatures relative to the hourly average temperature during the charging phase. A normalized ambient temperature ($NAT$) of 1 refers to an hourly average temperature equal to the charging period average. A NAT < 1 indicates a cooler-than-average charging period temperature, and NAT > 1 indicates a warmer-than-average temperature. In addition, the curves depicting solar elevation angle are overlaid to give a

sense of PV potential during the daily BTES system charge times. Symbols with a black field are charging months before the summer equinox, and those with a black fill are months after the equinox.

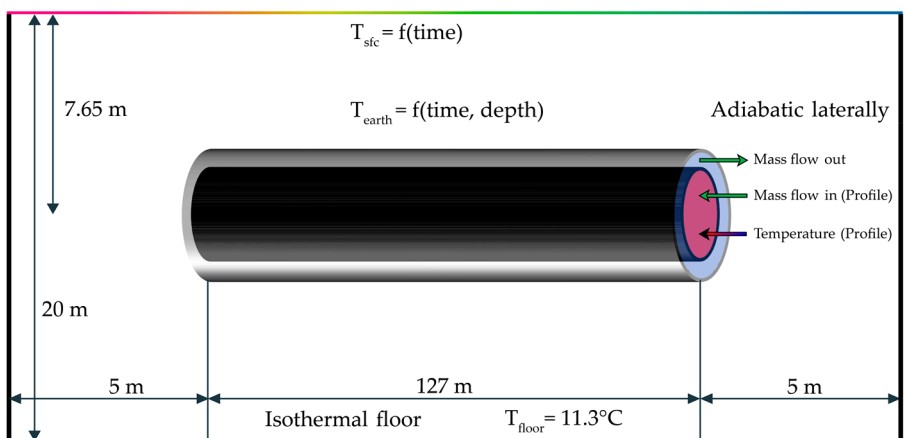

**Figure 8.** Flow schematic and dimensions for model validation.

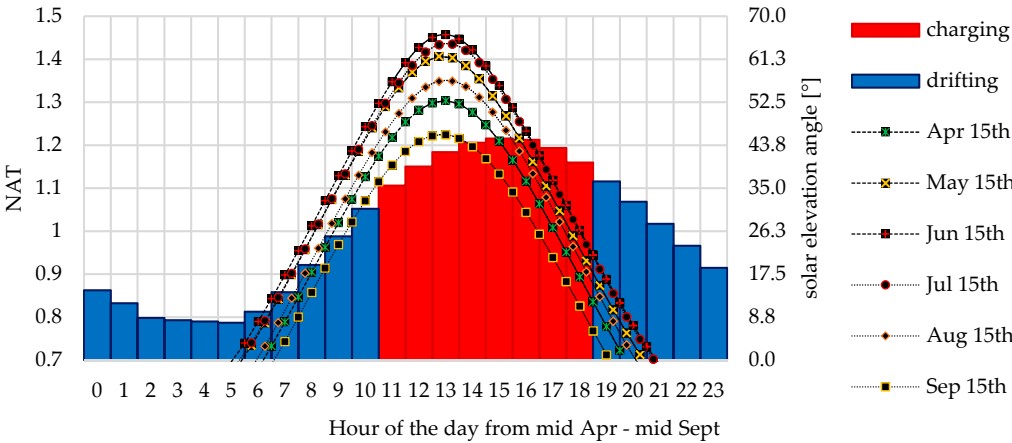

**Figure 9.** Normalized ambient temperature during the charging period.

The simplification of straightening the heated borehole (BOHR2) required an adjustment of the sensor positions. Vectors for linear segments defined by global positioning system (GPS) waypoints on BOHR2–BOHR6 and vectors for temperature sensors along each segment were used to determine sensor locations relative to the heated borehole in 3 dimensions. Measurement data in borehole Figure 10 below show how this process works in 2 dimensions.

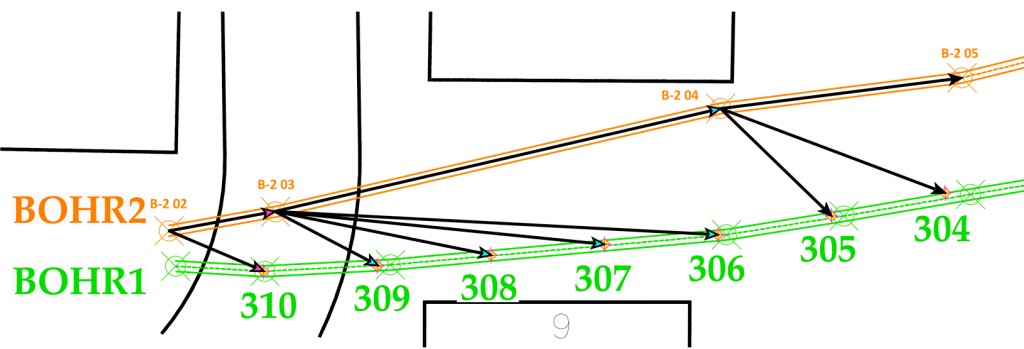

**Figure 10.** Vector analysis of temperature sensor location to heated borehole.

Heating (charging), drifting (no flow), and cooling (discharging) phases were executed during the experimental period. Part of the experimental period was used to validate the model for a period of 78 days between 10 May and 26 July.

The validated model was later expanded to 5 larger configurations of thermally active boreholes (4, 7, 12, 24, and 42) with two additional loading temperatures (40 °C, 50 °C) and an optimized flow rate. Figure 11 shows the layout of the 5 configurations.

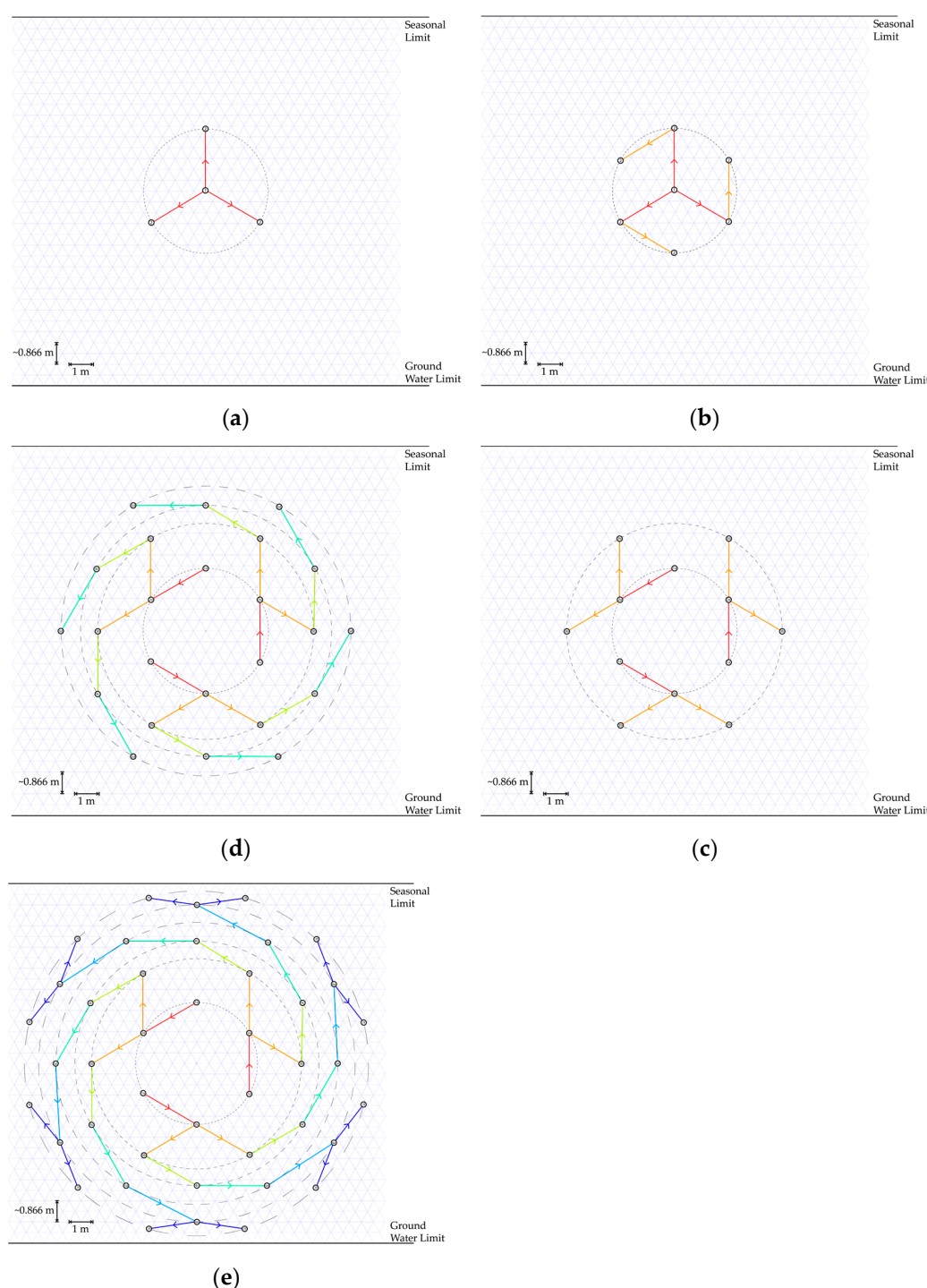

**Figure 11.** 4 (**a**), 7 (**b**), 12 (**c**), 24 (**d**), and 42 (**e**) 3D-HDD borehole configurations.

The center of each configuration was set to a depth of 20 m with a soil domain of 190 m in length, 40 m wide, and 40 m deep with a borehole length of 150 m. An optimized flow rate was determined by modeling a set of 5 different flow rates for each borehole

configuration. A flow rate maximizing BTES system thermal output and a flow rate maximizing BTES system coefficient of performance (COP$_{sys}$) were assessed (Section 2.4.). The flow rate maximizing BTES system thermal output was modeled for a 10-year period to assess a steady-state operating cycle representative of the average performance for a 50-year lifespan. The quantity of heat extracted was evaluated for the number of Swiss multifamily homes (MFH) supplied by a representative district heating network (Section 2.4.1). A basis of comparison between the number of boreholes installed, capacity of each configuration, and the influence of charging temperature was achieved through this line of analysis.

*2.4. BTES System*

The aboveground layout, intended use, and storage capacity are important choices for the overall design of the storage system. Type, power level, and quantity of energy required of a specific installation ultimately decide the temperature levels needed in the storage medium and overall volume of the storage [13].

Thermal efficiency ($\eta_{th}$) is dependent on the discharge temperature ($T_{out}$) and the return temperature ($T_{r,sto}$). Consequently, $\eta_{th}$, $T_{out}$, and $T_{r,sto}$ are dependent on at least the following characteristics of an energy storage system:

- the difference between the ambient temperature ($T_{amb}$) and storage temperature
- the difference between the return temperature from the heat sink ($T_{r,con}$) and $T_{amb}$
- the minimum and maximum outlet temperatures for a heat and cold storage, respectively
- the quantity of stored thermal energy ($Q_{in}$)
- the installed depth of the storage system
- the ratio between the length and cross-sectional width of the energy storage system
- the thermal properties of the underground, e.g., groundwater, permeability, anisotropic properties of different soil layers, belowground infrastructure, surface covering, etc.

Numerical models composed of 10 charging and discharging cycles were applied. The thermal performance on the 10th cycle (TP10) was chosen as the steady state operating point assuming charging, discharging, heat demand, and climate parameters were the same for every cycle [25]. TP10 was assumed to be the benchmark for evaluating thermal performance, equipment costs, and operation and maintenance costs for the operating life of the installation (50 a). The model generated return temperatures to the heat source ($T_{r,sto}$) during the charging phase and return temperature from the thermal energy storage ($T_{out}$) during the discharging phase. The first charging cycle from a chosen heat source would begin during the mid to late spring. A schedule for the annual operating cycle applied within this study began with a charging phase followed by a standby phase and ended with a discharging phase followed by a standby phase. The standby phases in early spring and autumn were 1 month each, while both charging and discharging phases were 5 months each. The quantities of stored and recovered thermal energy were determined using the generalized relationships in Equations (2) and (3).

$$Q_{in} = \dot{m}_{HTF} \cdot C_{p,HTF} \cdot \int \left(T_{in} - T_{r,sto}(t)\right) dt \tag{2}$$

$$Q_{out} = \dot{m}_{HTF} \cdot C_{p,HTF} \cdot \int \left(T_{out}(t) - T_{r,con}\right) dt \tag{3}$$

Thermal losses occur every cycle and are the difference between stored and recovered thermal energy. The thermal efficiency ($\eta_{th}$) for a BTES system is defined as [10,26,27]:

$$\eta_{th} = \frac{Q_{out}}{Q_{in}} \tag{4}$$

$Q_{in}$ represents the quantity of heat stored, and $Q_{out}$ represents the quantity of heat withdrawn from the storage. For the case of heat storage and cold storage, values of $\eta_{th} > 0.5$ and $\eta_{th} > 0.7$, respectively, are typical of a correctly dimensioned thermal energy storage system after several storage cycles [13].

Figure 12 expands the generalized thermal energy storage schematic in Figure 13 to the modeled BTES system in this study.

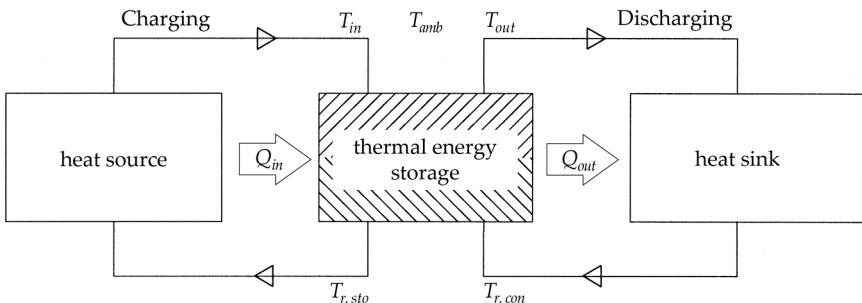

**Figure 12.** The BTES system modeled in this work.

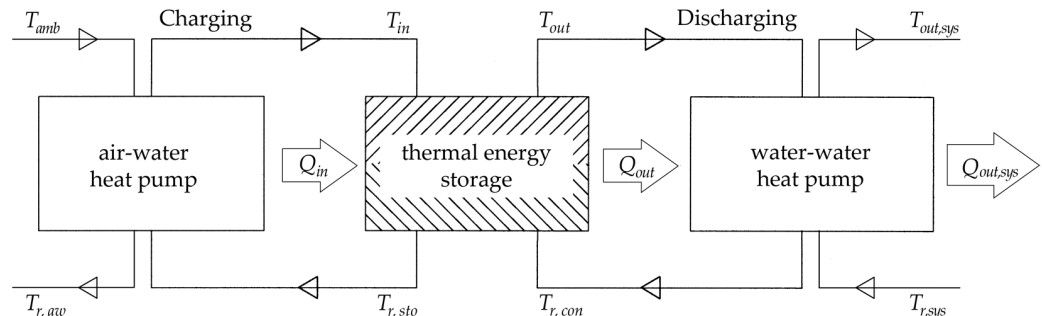

**Figure 13.** Temperature levels in a system with thermal energy storage.

Heat from summertime air $(T_{amb})$ is used to supply heat to the HTF with an air-water heat pump to charge the thermal energy storage $(Q_{in})$. The cooled fluid $(T_{r,sto})$ is returned to be heated. In winter, heat from the thermal energy storage is discharged $(Q_{out})$ at the outlet temperature $(T_{out})$. Discharged heat is increased with a water-water heat pump to supply a consumer with heat $(Q_{out,sys})$. The coefficient of performance of all heat pumps was evaluated using Equation (5) below.

$$COP_{(AW/WW)} = \phi \cdot \frac{T_{hi}}{(T_{hi} - T_{lo})} \tag{5}$$

In the case of the air-water heat pump, $T_{hi} \equiv T_{in}$ and $T_{lo} \equiv T_{amb}$ where $T_{in} = 40$, 50, and 60 °C. For the case of the water-water heat pump, $T_{hi} \equiv T_{out,sys}$ and $T_{lo} \equiv T_{out}$ where $T_{out,sys} = 60$ °C for domestic hot water, $T_{out,sys} = 35$ °C for building heat, and $T_{r,sys} = 25$ °C. In both cases, the quality grade of the heat pumps $(\phi)$ was assumed to be 0.5. The quantity of electrical energy needed to operate the air-water heat pump was determined as Equation (7) below.

$$Q_{el,\,AW} = \frac{\dot{m}_{HTF} \cdot C_{p,HTF} \cdot \int (T_{in} - T_{r,sto}(t))(T_{in} - T_{amb}(t))\,dt}{\phi \cdot T_{in}} \tag{6}$$

The discharging phase began on 15 October after a 1-month standby phase (drifting). Discharging was modeled as a continuous draw of heat from the BTES system for a 5-month period. Equation (7) below was used to determine the quantity of thermal energy extracted from the BTES system. It was assumed that the return temperature from the water-water heat pump $(T_{r,con})$ was a constant 10 °C, and that the temperature of the heat supplied to the system $(T_{out,sys})$ was a step function between 35 and 60 °C based on a model building

archetype (see Section 2.4.1.). The electrical energy needed to operate the water-water heat pump follows Equation (7) below.

$$Q_{el,WW} = \frac{Q_{out,sys}}{COP_{WW}} \tag{7}$$

The BTES system was not modeled to supply a specific quantity of heat. The amount of energy required to operate the water-water heat pump was a function of the thermal performance of the BTES system and the coefficient of performance of the heat pump itself.

Circulation of HTF to charge and discharge the BTES system is required for operation. The electrical energy required to pump the HTF through the BTES system was evaluated with Equation (8) where $\Delta p_l$ were the pressure losses in the coaxial pipe and the efficiency of the pump $(\eta_P)$ was set to 0.6.

$$Q_{el,hyd} = \frac{\dot{m}_{HTF}}{\rho_{HTF}} \cdot \frac{\Delta p_l}{\eta_P} \cdot t \tag{8}$$

The coefficient of performance for the BTES system $(COP_{sys})$ is defined as the amount of energy supplied by the BTES system to a customer or distribution network minus the electricity needed for operation divided by the amount of heat supplied to the BTES system as shown in Equation (9). Each configuration, including charging temperature, was evaluated using this method, assuming an average annual performance for the 10th year of operation.

$$COP_{sys} = \frac{Q_{out,sys}}{Q_{el,AW} + Q_{el,WW} + Q_{el,hyd}} \tag{9}$$

### 2.4.1. BTES System with a DHN

The building stock used in this study was based on a reference building for a MFH defined in [28]. The heating system for the reference building under the assumed conditions must provide 10 kW of heat, while each person is assumed to consume an average of 50 L of hot water at 60 °C per day [28–30]. From these boundary conditions, a thermal demand profile for the building was modeled. Figure 14 shows the calculated daily heat demand for a reference building [31–33].

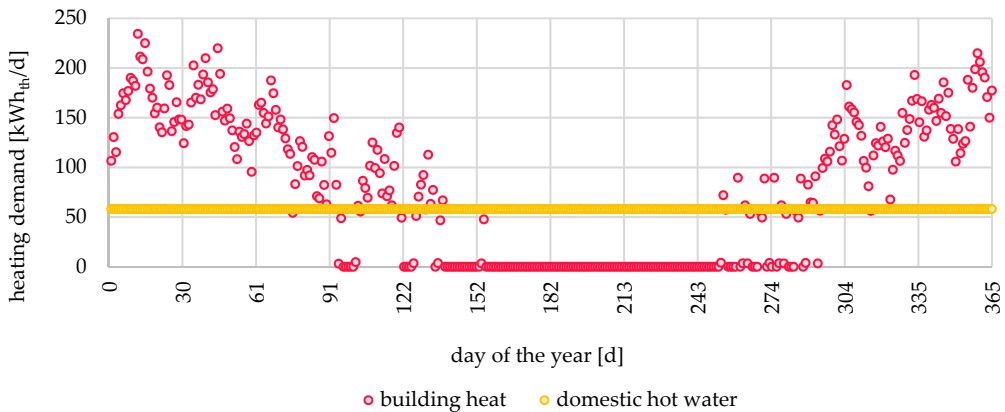

**Figure 14.** Annual demand profile of the reference multifamily house used in this work.

Buffer storage and equipment internal to each MFH was not considered in the modeling or cost calculation for this study.

The layout of a DHN with a BTES system in this study is described as a local (district or neighborhood) seasonal thermal energy store installed underneath existing infrastructure to supply building heat (BH) and domestic hot water (DHW). The storage itself takes no additional surface space overground. For the purposes of this study, an assumed layout of buildings was developed to model and calculate the economic performance of the system.

Figure 15 depicts the modeled layout of the DHN with a neighborhood of MFHs, BTES systems, heat pumps, and electrical connections. The clocks indicate the months where the BTES system is active (red), on standby (green), and inactive (white/blank). Each system was sized by the thermal output of each configuration $\left(Q_{out,sys}\right)$ and the quantity of MFHs this could serve. The MFHs were set up in rows to estimate the length of the DHS.

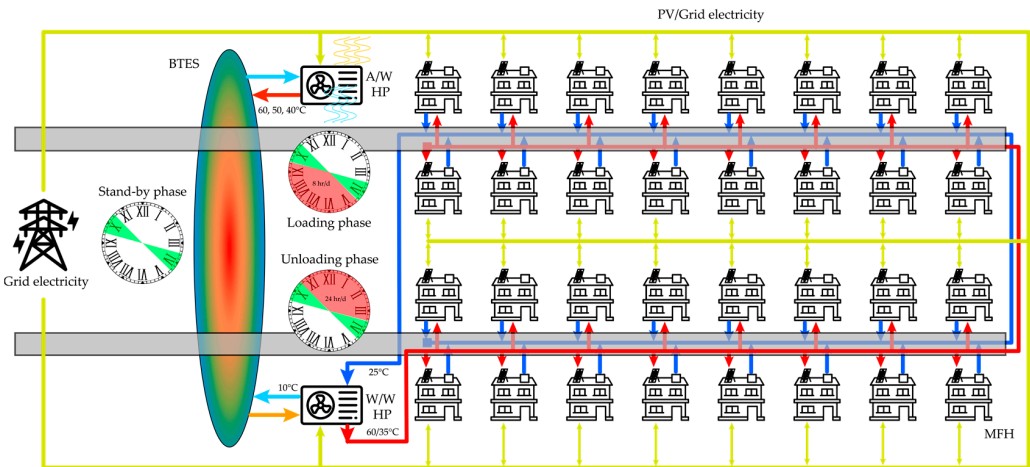

**Figure 15.** DHN with 3D-HDD BTES systems.

Assessment of a heating demand in given area is the linear heat density $(HD)$ defined by the amount of heat demand per unit linear distance (e.g., the length of pipe of the DHN). Equation (10) below was used to determine the linear heat demand density of the modeled neighborhood.

$$HD = \frac{2_{MFH}}{N_{MFH}} \cdot \frac{Q_{out,sys}}{\left(\sqrt{A_{MFH}} + sp_{MFH}\right)} \tag{10}$$

The assumed neighborhood in Figure 15 above had a linear heat density of $\approx 3100$ kWh/m. This exceeds the minimum of 1800 kWh/m recommended to ensure the economic feasibility of a DHN [34].

Given the linear heat density it is possible to calculate the diameter of the pipes necessary to supply the neighborhood with heat. Equation (11) below was used to determine the diameter of the DHN pipes $(\varnothing_{DHN})$ in m [35].

$$\varnothing_{DHN} = 0.0486 \cdot \ln(HD) + 0.0007 \tag{11}$$

### 2.5. BTES/DHN Equipment, Investment, and Operating and Maintenance Costs

The preceding methods and materials were used to calculate both the levelized cost of storage (LCOS) and the levelized cost of heat (LCOH). These calculations consider a discount rate of 5% [36], operating and maintenance costs, and the replacement of equipment during the lifetime of the storage system. The homogenous built environment was assumed to have no restrictions below ground level with other infrastructure, groundwater, etc.

A summary of energy flows as they relate to cost is shown in Figure 16.

Two system boundaries were considered for cost calculations. The inner boundary (light-dashed) surrounding the thermal storage determines the LCOS, while the outer boundary (heavy-dashed) surrounding all nodes determines the LCOH [37]. The LCOS can be calculated with the formula below where n represents the number of operating years, t the time, and r the discount rate (5%).

$$LCOS = \frac{I_c + \sum_1^n \frac{O\&M_C}{(1+r)^t}}{\sum_1^n \frac{E_3}{(1+r)^t}} \tag{12}$$

The LCOH was also calculated using a similar formula shown below.

$$LCOH = \frac{I_A + I_B + I_c + I_D + \sum_1^n \frac{O\&M_A + O\&M_B + O\&M_C + O\&M_D}{(1+r)^t}}{\sum_1^n \frac{E_3}{(1+r)^t}} \tag{13}$$

Data needed to assess costs are listed in the Cost Functions Table A1 in the Appendix A. Investment and O&M costs for each configuration and charging temperature are listed in Table A2.

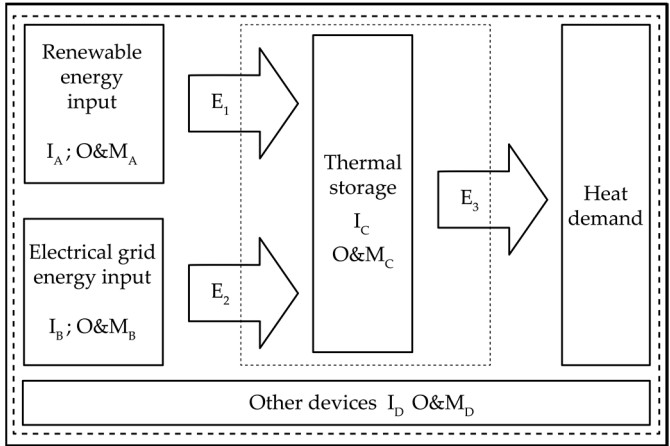

**Figure 16.** Schematic of energy flows.

### 2.5.1. Electrical Energy Costs

The electrical energy consumed to operate the circulation and heat pumps is a mix between photovoltaic (PV) solar panels and a connection to the electrical grid. It was assumed both sources of electricity were of equal cost per kWh ($C_{el}$) set to 0.21 CHF/kWh$_{el}$ [38].

### 2.5.2. Borehole Costs

Table 5 outlines costs for 3D-HDD construction and installation of piping in the dimensions assessed in this study.

**Table 5.** Borehole drilling and installation costs [39].

| Geology | up to 100 m | 100–200 m | Installation | Location Equip. |
|---|---|---|---|---|
| loose rock | 140 CHF/m | 120 CHF/m | 3000 CHF | 30 CHF/m |
| rock up to 100 MPa | 180 CHF/m | 170 CHF/m | 3000 CHF | 30 CHF/m |
| rock over 100 MPa | 230 CHF/m | 220 CHF/m | 3000 CHF | 30 CHF/m |

Investment and O&M costs for borehole configurations were modeled with cost Functions (1) and (2) [39].

### 2.5.3. Heat Pump Costs

Heat pumps are equipment which require replacement during the operating lifetime of the BTES system. This study assumed a replacement cycle once every 20 years for AWs and every 15 years for WWs.

The first cost tier of AWs is the class below 10 kW$_{th}$, with a second tier from 10 to 70 kW$_{th}$, as calculated with cost Functions (3) and (4), respectively. An absence of data between 70 and 500 kW$_{th}$ existed where linear interpolation bridged the price gap in this range. This was done by taking $I_{AW70}$. as the lower-bound and $\overline{I}_{AW500}$. as the upper-bound using cost Functions (4) and (5), respectively [40]. The resulting cost correlation between 70 and 500 kW$_{th}$ is given in cost Function (6). AWs larger than 500 kW$_{th}$ used a second linear

interpolation for average prices between 500 and 10,000 kW$_{th}$. This was calculated with cost Function (7) [40]. O&M costs of AWs were calculated with cost Function (8) [41].

Water-water heat pump investment costs were evaluated using a curve fit of data found for the capacity of the heat pump (WW$_{cap}$) in cost Function (9). O&M costs were assumed to be calculated the same way as air-water heat pumps with cost Function (10). Heat pump investment and O&M costs are listed below in Appendix A, Table A2, for each configuration and charging temperature.

### 2.5.4. Hydraulic Pump Costs

The investment and replacement costs of pumps were not included in this study. The cost of circulating the HTF through the boreholes was included in the borehole O&M cost.

### 2.5.5. DHN Costs

The cost of a district heating system is a function of HD. The investment cost of the piping needed to distribute heat to the consumers, as outlined in Section 2.4.1., is calculated with the cost Function (11) [35]. The length of the DHN was determined by the total number of MFHs for a given storage system, the length of the side of a building (assuming a square building), and the space between buildings defined in cost Function (12). The length of the DHN needed per MFH was around 15 m. O&M costs of the model DHN were calculated with cost Function (13) [35]. Thermal losses of the DHN were calculated with Equation (14) [34].

$$loss_{dhn} = 17 \cdot \sqrt{1000/HD} \tag{14}$$

The value used in this study was 10%.

## 3. Results

### 3.1. Experimental Results

Figure 17 shows the recorded inlet and outlet temperatures of the experimental borehole as well as the mass flow of the experiments conducted.

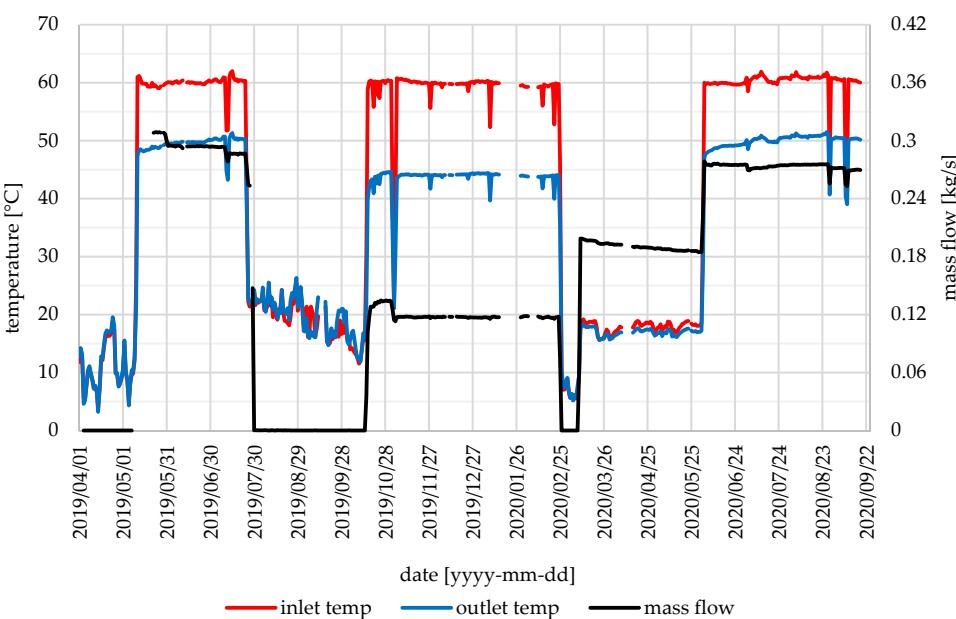

**Figure 17.** Inlet, outlet, and volume flow profiles of the experimental borehole BOHR2.

Temperature measurements were made inside the aboveground heating cell used to charge BOHR2. Temperatures recorded during zero flow conditions are representative of atmospheric conditions as the sensors were outdoors and aboveground.

Periods where data were not available or where logged values were unphysical are shown as gaps in the data in Figure 17. The first heating phase in spring 2019 was followed by a relaxation phase, a second heating phase, a brief second relaxation phase, a cooling phase, and finally a third heating phase in summer 2020.

The effects on temperature of these periods of operation are illustrated in the following figures. Each heatmap starts with four columns on the left indicating sensor ID and position (see Figure 4) followed by columns for orthogonal radial distance from BOHR2 (radius), depth from the surface (depth), and distance from the entry point into the ground to the sensor (length). The *x*-axis is time marked in 1-month intervals, and the *y*-axis values are the sequential temperature sensors ordered by entry point in BOHR1. Crosshatched areas indicate missing data. A color-coded temperature scale is set in a column to the right in °C where the top color is set to the maximum data value and the bottom color the minimum value. Figure 18 shows the recorded temperature data from BOHR2. It is clearly visible by the red areas where the first and third heating phases are located.

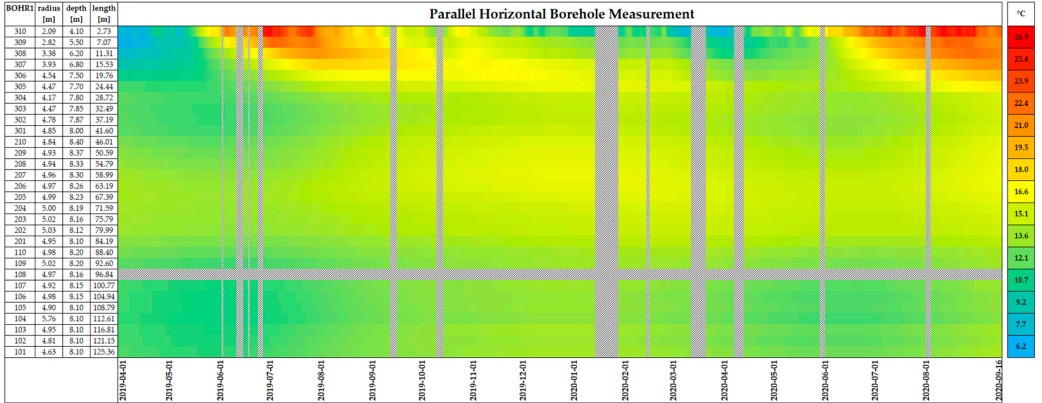

**Figure 18.** Parallel borehole temperature profile of BOHR2.

Figure 19 below shows the temperature recorded in BOHR3 and BOHR4 as they cross BOHR2 approximately 48 m from the entry point. BOHR3 crossed above the heated borehole, while BOHR4 crossed below.

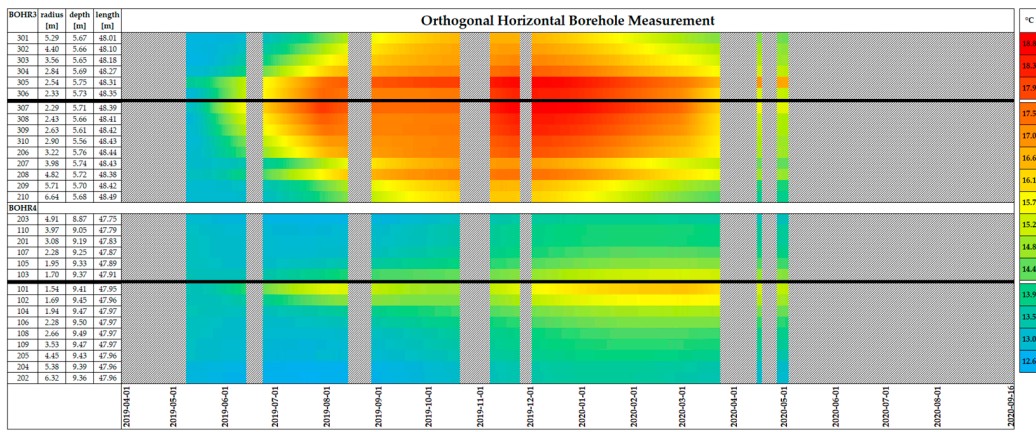

**Figure 19.** Orthogonal borehole temperature measurement ~48 m.

Figure 20 below shows the temperature recorded in BOHR5 and BOHR6 as they cross BOHR2 approximately 109 m from entry point. BOHR5 crossed above the heated borehole, while BOHR6 crossed below.

Three additional vertical boreholes were drilled to allow for a vertical profile of temperature measurements. The layout of each plot is formatted the same as the previous plots with the addition of an expanded view for the date range of the recorded data.

Figure 21 below shows the vertical temperature profile for borehole KERN1. The data recording began 10 February 2020 just before the cooling phase, and ends 17 April 2020, capturing a part of the cooling phase that began 10 March 2020.

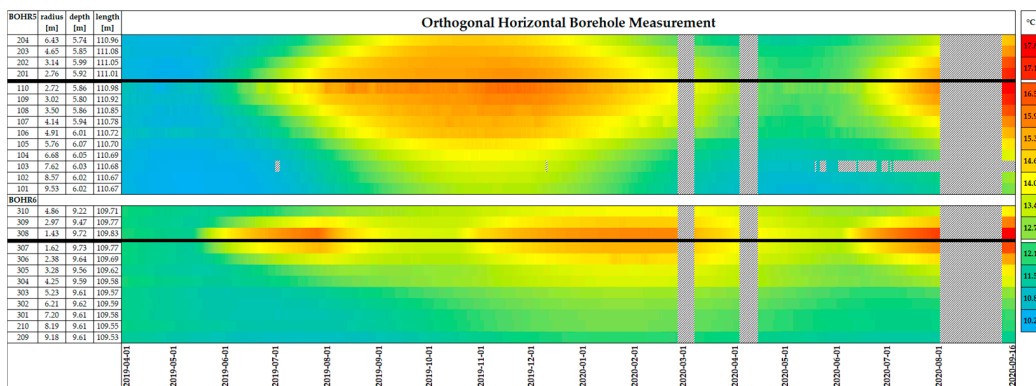

**Figure 20.** Orthogonal borehole temperature measurement ~109 m.

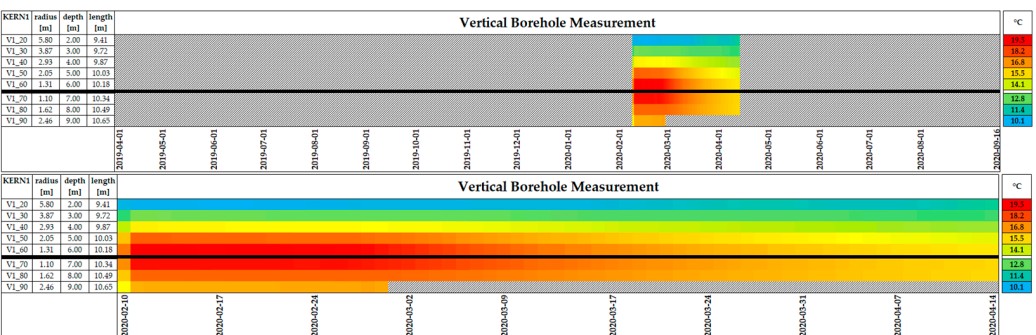

**Figure 21.** KERN1 vertical borehole temperature profile.

Figure 22 below shows the vertical temperature profile for borehole KERN2. The data recording began 10 February 2020 just before the cooling phase, and extends to the end of the measurement period, capturing the third heating phase and some seasonal effects.

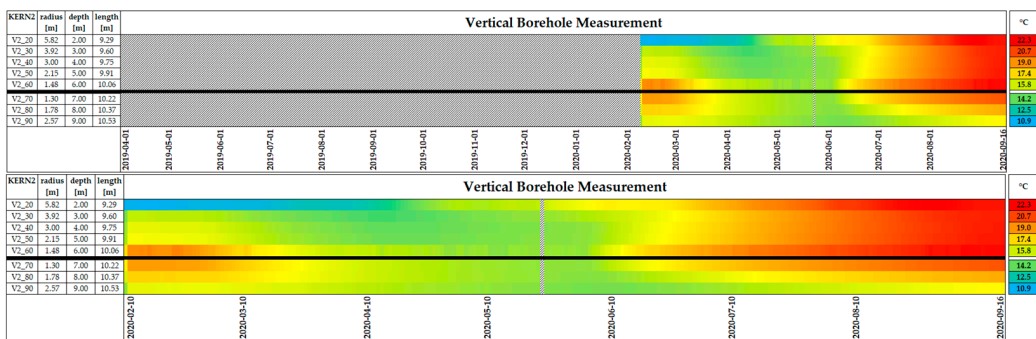

**Figure 22.** KERN2 vertical borehole temperature profile.

Figure 23 below shows the vertical temperature profile for borehole KERN3. The data recording began 10 February 2020 just before the cooling phase, and ends 17 April 2020, capturing a part of the cooling phase that began 10 March 2020.

### 3.2. Model Validation

Measurements were conducted to allow validation of the single-bore model. The first heating period was used. The validation of the model was first evaluated on the

thermal performance of the inlet and outlet temperatures of the HTF in the heating cell and from the temperature sensors in the measurement boreholes. In situ experimental temperature measurements were compared with modeled values at the corresponding sensor locations for a 78-day period which began on 10 May 2019 and ended on 26 July 2019. Figure 24 shows both modeled and measurement values for inlet and outlet temperatures (left axis) as well as the difference between model and measurements (right axis) during the validation period.

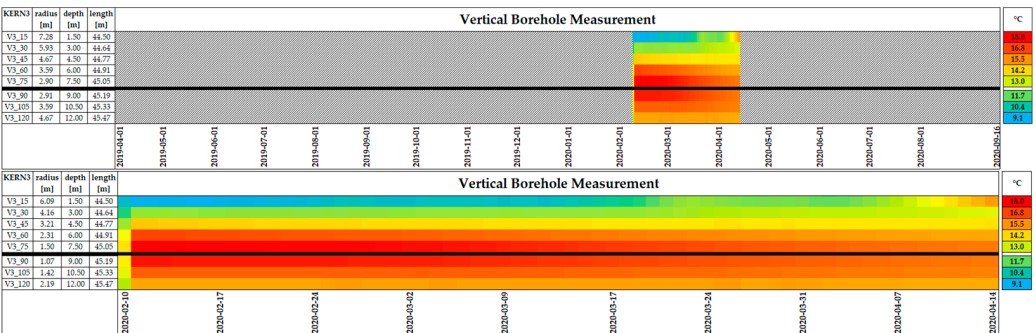

**Figure 23.** KERN3 vertical borehole temperature profile.

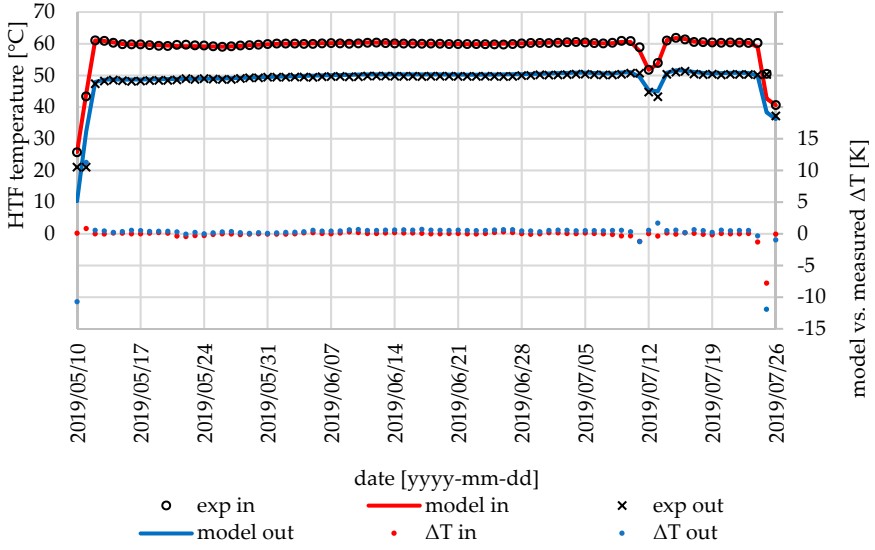

**Figure 24.** Comparison of model data and measurements for experimental borehole inlet and outlet temperatures.

The experimental measurements collected and mapped during the validation period (see Section 3.1.) were compared with the modeled results. Twenty-nine individual sensors recorded measurements in BOHR1 (see green line Figure 4). Figure 25 below shows a heatmap of model output against measured temperature for the positions given for each sensor. Each sensor is coded with a three-digit ID, its radial distance from the borehole, depth from ground level, and length from entry point. An offset color scale was set to highlight features with a value for model/measure agreement equal to zero.

For more specific analysis, two sets of two orthogonal measurement boreholes were located at ~48 and ~109 m from the entry point of the heated borehole. BOHR3 to BOHR6 were analyzed for modeled temperature versus measured temperature values (see blue, red, magenta, and cyan lines Figure 4). Figure 26 below shows the first set (BOHR3 and BOHR4) of sensors and where they cross the heated borehole. BOHR3 crosses ~2.3 m above the heated borehole, and BOHR7 crosses ~1.6 m below the heated borehole. The black

lines in the figures indicate the approximate position of the heated borehole relative to the sensors.

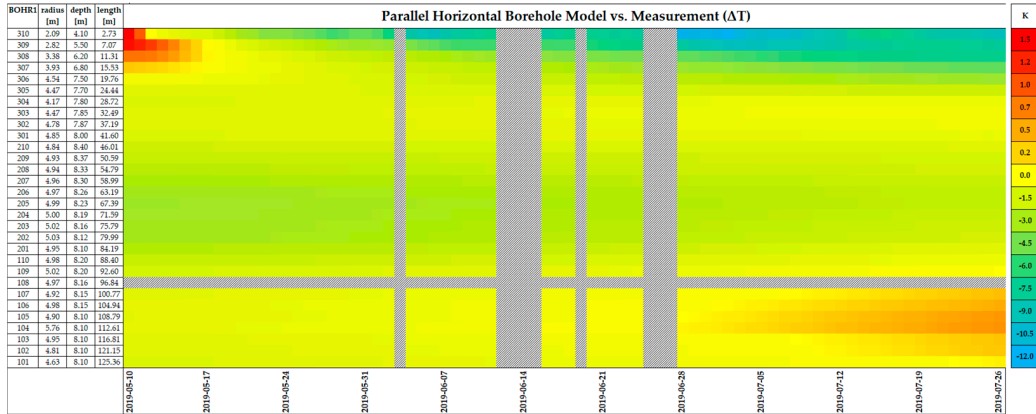

**Figure 25.** Model performance vs. measured values of temperature during 78–day charging period.

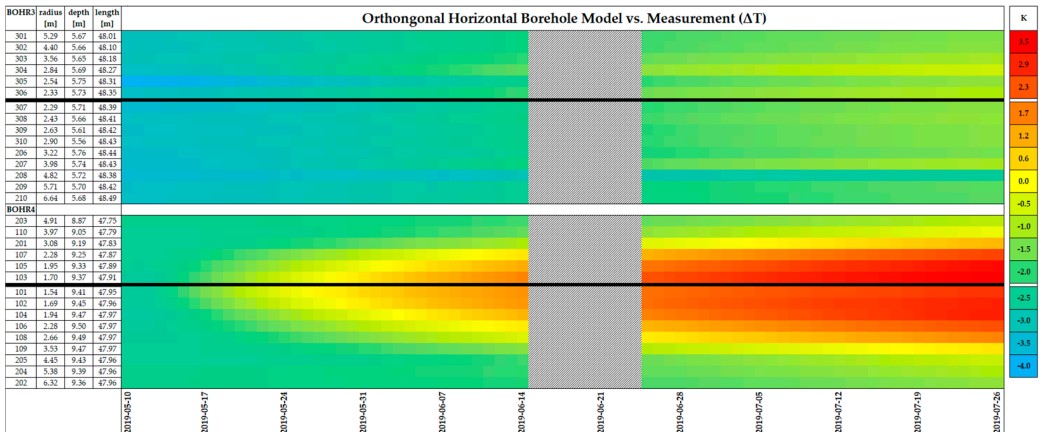

**Figure 26.** BOHR3 and BOHR4 modeled temperatures plotted against measured temperatures.

Figure 27 below shows the second set (BOHR5 and BOHR6) of sensors and where they cross the heated borehole. BOHR5 crosses ~2.7 m above the heated borehole, and BOHR6 crosses ~1.5 m below the heated borehole.

Thermal performance of the experimental borehole was compared to the model by evaluating energy balance. The quantity of energy loaded was determined using Equation (2) for both sets of modeled and empirical data. Comparison of the experimental data and the model revealed a charging heat deficit of 3.3% as predicted with the model.

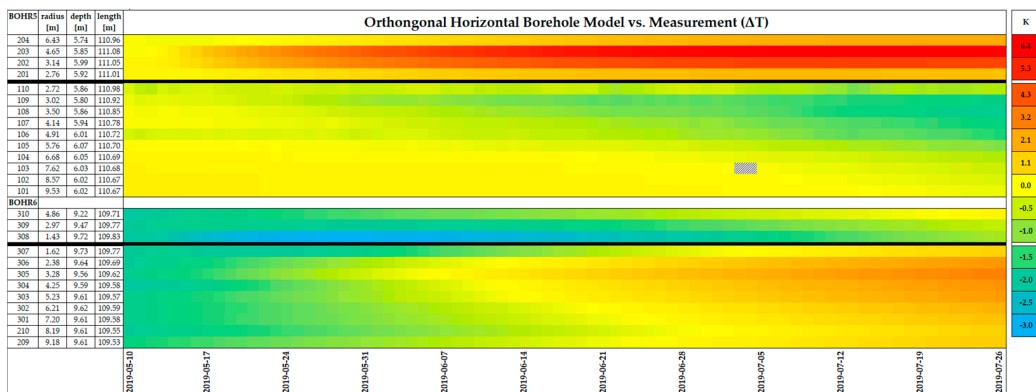

**Figure 27.** BOHR5 and BOHR6 modeled temperatures plotted against measured temperatures.

### 3.3. Multi-Borehole Configurations

The validated borehole model was expanded for thermal analysis of larger borehole fields. Configurations of 4, 7, 12, 24, and 42 boreholes were modeled and analyzed as shown in Figure 12. A set of five flow rates were modeled to determine the optima with polynomial curve fits. The maximum net supplied heat and maximum seasonal performance factor of the storage system for each borehole configuration and charging temperature were determined using a single cycle model (1a). The quantity of thermal energy discharged by the storage system is $(Q_{out,sys})$. The net thermal energy $(Q_{net})$ is the thermal energy output of the storage system minus the electrical energy needed to operate the storage system shown in Equation (15). The thermal ratio $(Q_r)$. is the ratio of net thermal energy discharged by the storage system to the charged thermal energy shown in Equation (16) below.

$$Q_{net} = Q_{out,sys} - \left( Q_{el,pump} + Q_{el,AW} + Q_{el,WW} \right) \tag{15}$$

$$Q_r = \frac{Q_{net}}{Q_{in}} \tag{16}$$

Figure 28 shows the net supplied heat $(Q_{net})$ and the thermal ratio $(Q_r)$. of the storage system for the assessed flow rates of the 42-borehole configuration charged at 60 °C. Green dashed lines indicate the flow rate (*x*-axis), net discharged energy (left *y*-axis), and thermal ratio $(Q_r)$ (right *y*-axis) that maximized thermal output. Yellow dashed lines indicate the same quantities for the flow rate optimizing the thermal ratio.

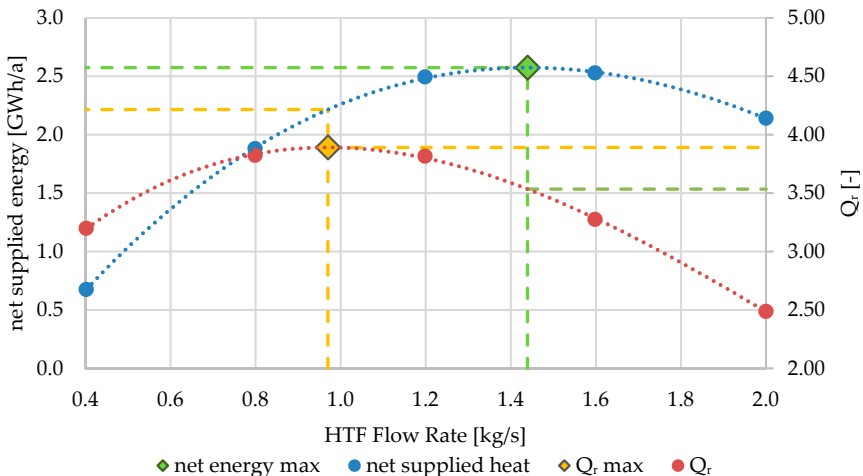

**Figure 28.** Modeled flow rates vs. net supplied energy for 42 boreholes charged at 60 °C.

A summary of modeled flow rates for different combinations of configurations and charging temperatures is shown in Figure 29. The bar plot represents the thermal output $(Q_{out,sys})$ of the storage system and the dotted lines the thermal ratio $(Q_r)$. for each configuration and charging temperature. The color codes differentiate between the flow rate for maximum energy (blue) and the flow rate for maximum thermal ratio (green).

For each configuration and charging temperature, the choice was made to choose the flow rate which maximized the thermal output of the system (blue) over the flow rate which maximized the thermal ratio (green). The flow rate maximizing energy output for each combination of configuration and charging temperature was modeled for a 10-cycle period. Figure 30 below shows the thermal efficiency (Equation (2)) of each configuration by number of boreholes and charging temperature (left *y*-axis) and system COP (right *y*-axis).

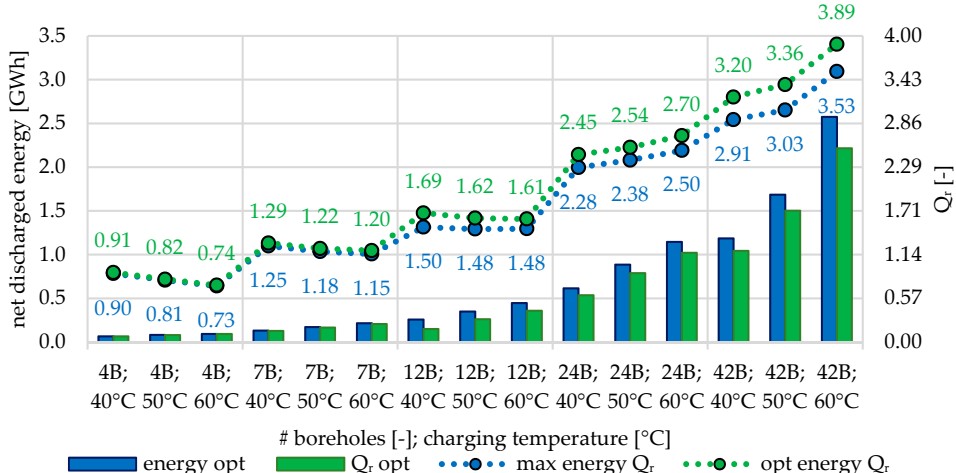

**Figure 29.** Thermal and storage system performance at the two optimized flow rates.

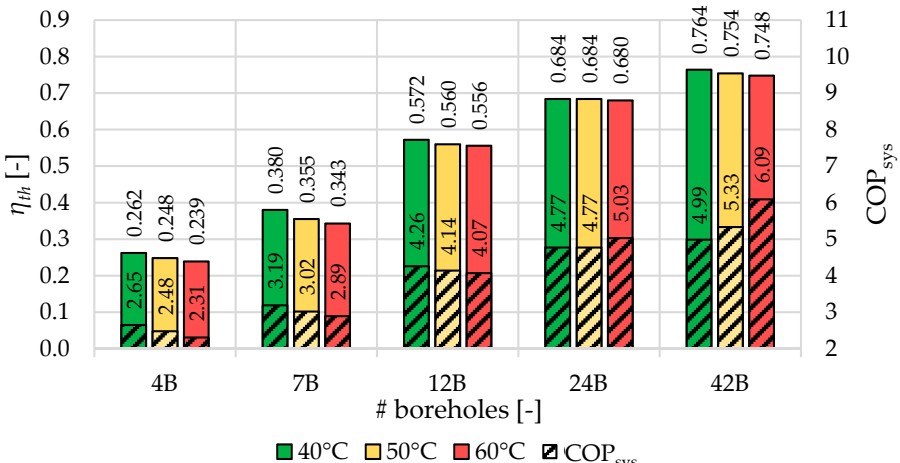

**Figure 30.** Thermal efficiency by charging temperature and configuration.

Each modeled configuration and charging temperature were tightly grouped by borehole configuration and trend toward a maximal value of ~0.8.

A final look at thermal performance considers the total annual heat output (left *y*-axis) and specific heat output $\left(Q_{out,sys}/\left(N_{bh} \cdot L_{bh}\right)\right)$. (right *y*-axis) of each configuration and charging temperature in Figure 31. The specific heat output is the thermal output of the system normalized by the total length of the configuration.

### 3.4. Economic Evaluation

The economic assessment of the storage system was evaluated by applying the cost data (Section 2.5.) to the thermal performance of each configuration and charging temperature to calculate the LCOS and LCOH. Figure 32 below shows the LCOS.

The LCOS only includes the investment cost of the BTES system and the operation and maintenance of the boreholes. The LCOH includes the costs and losses of a district heating network for heat delivery to the modeled buildings including all other equipment and energy costs. Figure 33 below shows the LCOH for each configuration modeled (2.3.) and the annual thermal energy delivered by the district heating network as calculated with the 10th year storage system performance over a 50-year operation lifetime.

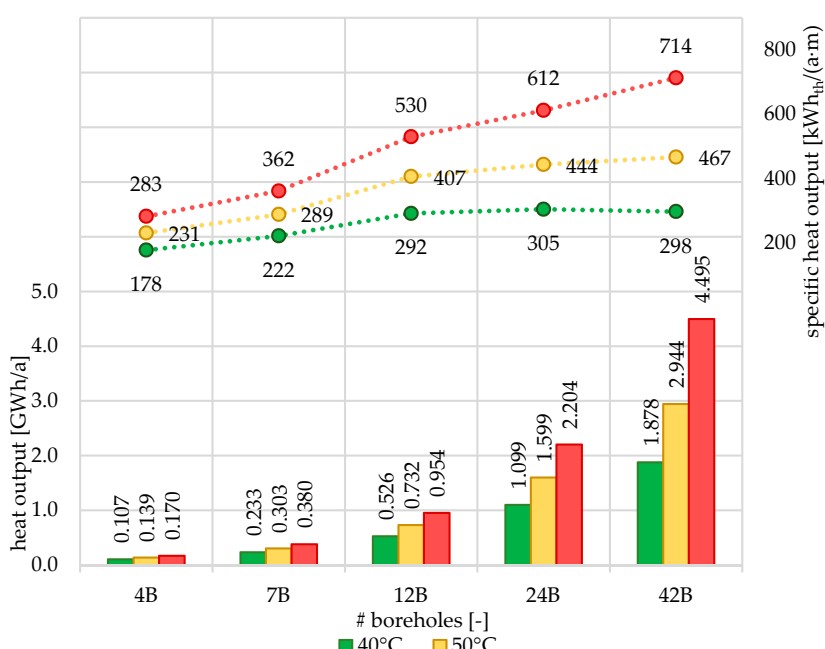

**Figure 31.** Specific heat output (top) and annual heat output (bottom) by configuration and charging temperature.

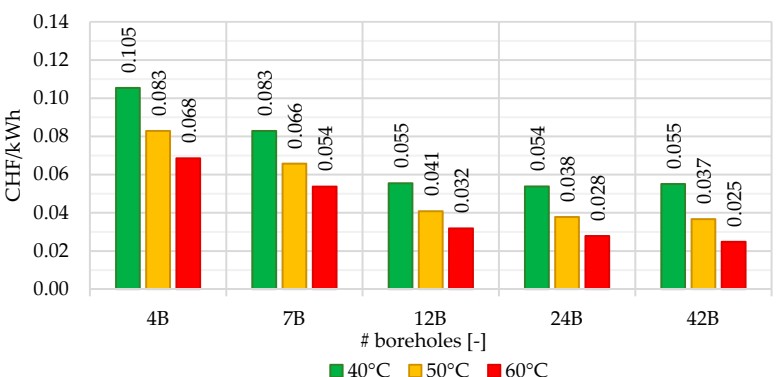

**Figure 32.** LCOS for each configuration and charging temperature.

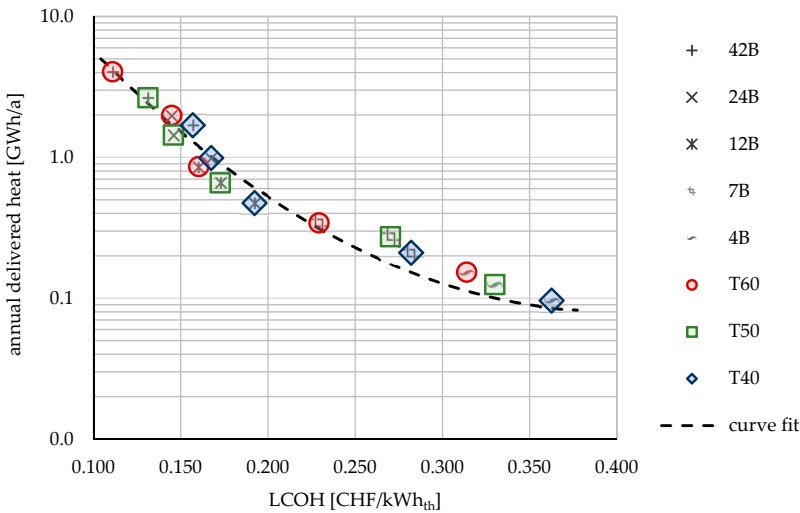

**Figure 33.** LCOH and annual delivered heat for each configuration and loading temperature.

Figure 34 shows the investment cost of the boreholes, charging heat pump, discharging heat pump, and district heating network for each configuration and charging temperature.

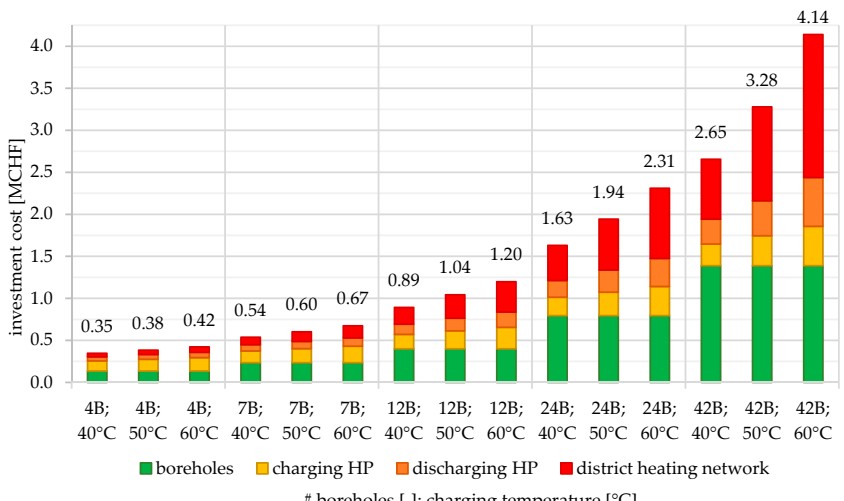

**Figure 34.** Investment cost for each modeled configuration and charging temperature.

The color-coded bars represent the proportion of the total investment cost for each configuration and charging temperature modeled. The borehole installation and district heating network bear the largest proportion of the investment cost.

## 4. Discussion

### 4.1. Experimental Results

Experimental measurements of temperature showed the influences of borehole depth, surface coverage, and belowground features which were not included in the models. A charging phase was initiated during winter when a discharging phase should have been started. The first charging phase was used for model validation as it represented the only period consistent with a seasonal BTES system operating cycle. Comparison of measurements made during a 30-day period before the first charging phase and the modeled output showed temperature deviations in the entry length region (see Figure 5, BOHR1, 310–306) of BOHR1 consistent with weather conditions at the surface. This is illustrated by the absence of a warm temperature signal for the same set of sensors during the second heating phase (Figure 5, Table 4) where the loaded heat is lost to the influence of cold-season temperatures at the surface. This did not produce a significant difference in the quantity of heat charged during the validation period with a single borehole model (3.3%). However, configurations with many boreholes would be exposed to shallow depths and potential losses in the entry length region. Configurations with multiple boreholes would need to be drilled from a belowground starting point (e.g., a basement), below a surface structure, or into a steep slope to buffer the effects of seasonal surface variation. Configuration radius for very large configurations would also dictate the targeted installation depth to minimize thermal interaction with the surface and optimize thermal performance.

BOHR1 passed near buildings, under a driveway, below a warehouse, next to an insulated belowground floor, and next to a ramp. Comparing the first month (1 April–1 May 2019) of recorded data before any heat had been added or withdrawn from the underground helps to understand the influences and differences between model and measurement. Overall, the model performed well considering it did not include any of the complex nature of the structures the test-site. Sensors 310–307 revealed both warmer and cooler regions during the month prior to the first heating phase. The deviations from the model shown above correspond to difference from the average conditions for 1991–2021 and conditions for 2019. Sensors recorded temperatures originating from the weather for

the period between mid-November 2018 and the beginning of January 2019. Warmer-than-average temperatures were recorded at the end of December, which corresponds well with the peak in temperatures on 18 April 2019. Likewise, preceding cold periods are reflected in the recorded data at sensor 310. Soil in the deep solar zone has less sensitivity to short-term weather events at the surface. Sensor 309 displayed weather with an approximate lag of ~3.5–5 months and captured colder than normal weather during this period. Between 1 October 2018, and 15 December 2018, 57% of the temperatures recorded were below-average, with the average temperature for October 2018 being 1.3 °C below the average. Sensors 308 and 307 are the last near-surface sensors with a natural surface, which displayed the effects of weather with even greater dampening of surface events. Sensors 306–203 were below an artificial structure. As previously mentioned in Sections 2.1 and 2.2, a covered or sealed surface acts as a buffer between conditions underground and the ambient conditions aboveground. In addition, sensors 203–110 are along a section of the building where there is an insulated belowground floor as shown in Section 2.2. This further contributed to the volume of soil that was nearly 3 K warmer than the modeled soil in the same physical location. The remaining measurement sensors in BOHR1 (109–101) range from ~0.4 K to ~1.1 K warmer than the modeled soil. A depth of 8.1 m lags ambient temperature conditions by ~5.5–6.5 months. Weather data from mid-September to mid-October 2018 shows 65% of data points were below-average for this period, which correlates well with measured temperatures being cooler than the model at this depth.

Temperature measurements made in BOHR1 reflect the various phases of heating, drift, and cooling. H1 and H3 phases are well-presented in the measured data, while the second did not show a similar temperature signature. This is due to heating during winter, where cooler temperatures are reaching sensors 310 and 309. Some heat is still contained within the soil below a depth of 6 m, but not with the intensity of H1 and H3 phases. BOHR3 and BOHR4 recorded temperatures during just before H1 to just past H2 phases (Figure 19), whereas BOHR5 and BOHR6 recorded temperatures during all phases (Figure 20). An asymmetric heating pattern was observed with more heat reaching sensors in BOHR3 and BOHR5 than in BOHR4 and BOHR6, respectively, in all heating phases recorded. It is unknown why such a stratification occurred. All three vertical boreholes (KERN1–KERN3) recorded small samples of data relative to the overall operation period. The data shown in Figures 21–23 are consistent with the heating, drifting, and cooling phases during the operational period.

Comparison of measurements made during the validation period with the single-bore model output showed a difference of 1.3–3 K higher underneath the warehouse (see Figure 4, BOHR1, 305–109). This was expected due to the insulating effects of concrete slabs covering the surface over BOHR1 sensors 305–301, 202–109, and the warehouse covering sensors 210–203. Additionally, BOHR3 and BOHR7 were covered by the warehouse and showed mixed results with a difference between −1.5–3.5 K. This occurred near the edge of the building, where disturbed soil is often found and there is a path for increased rainwater to flow from the roof. The actual causes for the deviations are not known.

A comparison of measurements and validation model output from sensors near the basement and access ramp showed the largest deviations (−3.0–6.7 K). The coldest measurement is theorized to pass a region of extremely disturbed soil (increased thermal diffusivity) at reduced depth below the ramp (BOHR6, 308). The hottest measurement occurred in BOHR5 (203 and 202) in a region that lay at an even shallower depth below the ramp in extremely disturbed soil.

*4.2. Model Output and Economic Analysis*

Analysis of thermal efficiency and system COP (Figure 33) revealed two trends, the first of which was a decrease in thermal efficiency with an increase in charging temperature for each configuration. This was expected as the difference between charging temperature and ambient would dictate not only energy density of the storage, but also the losses to the environment. However, configurations with more than 27 boreholes with optimal spacing

and storage volume can be dimensioned to minimize losses. Higher loading temperatures paired with larger configurations can increase thermal efficiency and system COP over smaller configurations at the same loading temperatures. Higher storage temperatures require less energy to lift the stored heat to temperature levels for use (building heat and domestic hot water). The loss in thermal efficiency was small compared to the savings of energy needed to raise the temperature of stored heat for use.

The relationship between a configuration's size, thermal efficiency, and system COP is the opposite for configurations with less than 27 boreholes. It was observed that lower loading temperatures can be used (>35 °C) with a minimum of 12 boreholes to maintain a thermal efficiency > 50%.

Analysis of LCOS, LCOH, and investment cost revealed reduced costs per $kWh_{th}$ with scale. The trend in LCOS in Figure 32 shows decreasing cost per kWh with increasing configuration size and charging temperature with a minimum of 0.025 $CHF/kWh_{th}$. The trend in Figure 33 shows a decreasing LCOH for heat delivered to the building, including distribution costs and losses with size and charging temperature. Larger configurations with higher charging temperatures incurred more investment costs to commission a system as shown in Figure 34. Results from multi-bore numerical models suggest the combination of a charging temperature $\geq 60$ °C and storage with >42 boreholes has the potential to break below an LCOH of 0.10 $CHF/kWh_{th}$. Equipment costs can be further reduced by raising the storage temperature to a level where extracted heat from the BTES system renders the discharging heat pump unnecessary. Additional savings are possible in a district with higher linear head demand density reducing the length of the district heating network.

## 5. Conclusions

The use of 3-dimensional, horizontally directed drilling of boreholes is a viable technology for installing arrays of boreholes for the purpose of seasonal thermal energy storage. This drilling technique allows access to storage sites that are not possible to reach with vertical borehole drilling. It has the potential to transform the way excess energy production in summer from renewable sources such as solar–thermal or photovoltaic panels can be utilized in winter in locations with surfaces which cannot be altered. It can reduce costs associated with a district heating network by being placed closer to intense demand centers by reducing pipe and trench length. Although slightly more expensive than vertical borehole drilling per meter, this would be offset by reduced district heating network length and provide savings by horizontally drilling the district heating network as well. In addition, artificial surface cover has an underground warming effect by insulating the underground from the cold seasons and nighttime temperatures. Furthermore, a BTES system located underneath a surface or structure requiring low temperature heat would benefit from storage heat losses.

Measurements made using a network of sensors around a heated borehole in a test site were in good agreement with the numerical model developed in this work. The trend of the results points to expanding configurations to include more boreholes and, where possible, increasing the charging temperature to further reduce lifting temperatures during discharging and increase the potential for direct use, thus eliminating the need for a discharging heat pump.

It is imperative more research with specific studies incorporating the most relevant seasonal thermal energy storage applications is investigated along with sector coupling and thermal grids. Optimizing configuration geometry for a specific heat demand and temperature level to minimize heat losses is essential for the construction of a pilot and demonstration. Models incorporating surface coverage (e.g., buildings) are a must for accurately assessing BTES systems placed below structures. A case representing an existing dense urban center without an existing thermal energy storage nor district heating network, as well as a case representing a very intense consumer of thermal energy (e.g., an airport) where no district heating network is necessary, would cover the highest linear heating densities with and without a district heating network and thus lower the LCOH. Extension

of system operation to store cold underground in winter for use in summer would further cover the heating and cooling demand of a consumer. The utilization of extracted cold temperatures during summer would preheat return temperatures to the heat source, thus reducing charging costs.

**Author Contributions:** Conceptualization, L.F.; methodology, W.V.; validation, R.B. and S.A.; formal analysis, R.B.; investigation, R.B.; writing—original draft preparation, R.B.; writing—review and editing, L.F., W.V. and R.B. All authors have read and agreed to the published version of the manuscript.

**Funding:** The authors gratefully acknowledge Innosuisse and Schenk AG for financial contributions to this work under the project Seasonal 3D Geothermal Heat Storage with Horizontally Directed Drilling (project ID 27935.1 PFIW-IW). This research is part of the activities of the Swiss Competence Center for Energy Research on Heat and Electricity Storage (SCCER HaE), which was financially supported by the Swiss Innovation Agency–Innosuisse. This research was also financially supported by the Swiss Federal Office of Energy as part of the SWEET PATHFNDR project.

**Data Availability Statement:** Not applicable.

**Conflicts of Interest:** The authors declare no conflict of interest.

## Nomenclature

| Term | Definition | Unit |
|------|-----------|------|
| 3D-HDD | 3-dimensional horizontally-directed drilling | - |
| AW | air-water heat pump | - |
| BH | building heat | - |
| BOHRX | horizontal borehole X | - |
| BTES | borehole thermal energy storage | - |
| $COP_i$ | coefficient of performance of i | - |
| DHN | district heating network | - |
| DHW | domestic hot water | - |
| GPS | global positioning system | - |
| HDPE | high density polyethylene | - |
| HTF | heat transfer fluid | - |
| IEA | International Energy Agency | - |
| IGE | Institute für Gebäudetechnik und Energie; HSLU | - |
| KERNX | vertical borehole X | - |
| MFH | Swiss multi-family house | - |
| $N_i$ | number of component i | - |
| PV | photovoltaic | - |
| SFOE | Swiss Federal Office of Energy | - |
| WW | water-water heat pump | - |
| # | number | - |
| bh | borehole | - |
| calc | calculated | - |
| dhn | district heating network | - |
| el | electric | - |
| hyd | hydraulic | - |
| lat | latitude | - |
| long | longitude | - |
| r | discount rate | - |
| recom | recommended | - |
| th | thermal | - |
| yr | year | - |
| $A_i$ | area of component i | $m^2$ |
| $C_{p,v}$ | volumetric specific heat capacity | $MJ/m^3$ |
| $C_i$ | cost of component i | CHF/kWh |
| $\varnothing_i$ | diameter of component i | m |
| $E_i$ | energy of component i | kWh |

| Term | Definition | Unit |
|------|-----------|------|
| HD | linear heat demand density | kWh/m |
| $\bar{I}_i$ | average investment cost of component i | CHF |
| $I_i$ | investment cost of component i | CHF |
| λ | thermal conductivity | W/(m·K) |
| LCOH | levelized cost of heat | CHF/kWh$_{th}$ |
| LCOS | levelized cost of storage | CHF/kWh$_{th}$ |
| $L_i$ | length of component i | m |
| O&M$_i$ | operating and maintenance cost of component i | CHF/year |
| $Q_i$ | energy quantity of i | kWh |
| ρ | density | kg/m$^3$ |
| sp$_i$ | spacing between component i | m |
| t | operating years | years |
| XX$_\#$ | X-X heat pump capacity | kW |

## Appendix A

**Table A1.** Cost functions.

| Term | Equation | Unit | Index |
|------|----------|------|-------|
| $I_{bh}$ | $220 \cdot L_{bh} \cdot N_{bh} + 3000$ | CHF | (1) |
| $O\&M_{bh}$ | $0.01 \cdot I_{bh} + Q_{el,hyd} \cdot C_{el}$ | CHF/year | (2) |
| $I_{AW<10}$· | $1167 \cdot AW_{<10} + 19618$ | CHF | (3) |
| $I_{10 \leq AW \leq 70}$· | $1167 \cdot AW_{10 \leq | \leq 70} + 19948$ | CHF | (4) |
| $\bar{I}_{AW500}$· | $615 \cdot \bar{AW}_{500}$ | CHF | (5) |
| $I_{70 < AW < 500}$ | $I_{AW70} + \frac{(\bar{I}_{AW500} - I_{AW70})}{(AW_{500} - AW_{70})} \cdot \left( AW_{70<|<500} - AW_{70} \right)$· | CHF | (6) |
| $I_{AW>500}$ | $\bar{I}_{AW500} +$ $\left[ \frac{\bar{I}_{AW500}}{AW_{500}} - \frac{\left( \frac{\bar{I}_{AW500}}{AW_{500}} - \frac{\bar{I}_{AW10000}}{AW_{10000}} \right)}{(AW_{10000} - AW_{500})} \cdot AW_{\geq 500} \right] \cdot$ $(AW_{\geq 500} - AW_{500})$ | CHF | (7) |
| $O\&M_{AW}$ | $0.01 \cdot I_{AW} + Q_{el,AW} \cdot C_{el}$ | CHF/year | (8) |
| $I_{WW}$ | $WW_{cap} \cdot \left[ -0.229 + 0.355 \cdot WW_{cap}^{0.192} \right]^{(-1/0.192)}$ | CHF | (9) |
| $O\&M_{WW}$ | $0.01 \cdot I_{WW} + Q_{el,WW} \cdot C_{el}$ | CHF/year | (10) |
| $I_{dhn}$ | $[315 + 2225 \cdot (0.0486 \cdot \ln(HD) + 0.0007] \cdot L_{dhn}$ | CHF | (11) |
| $L_{dhn}$ | $\frac{N_{MFH}}{2} \cdot \left( \sqrt{A_{MFH}} + sp_{MFH} \right)$ | m | (12) |
| $O\&M_{dhn}$ | $0.002295 \cdot Q_{dhn}$ | CHF | (13) |

**Table A2.** Investment and O&M costs for the modeled systems.

| Boreholes $T_{in}$ | $I_{bh}$ [CHF] | $O\&M_{bh}$ [CHF/yr] | $I_{AW}$ [CHF] | $O\&M_{AW}$ [CHF/yr] | $I_{WW}$ [CHF] | $O\&M_{WW}$ [CHF/yr] | $I_{dhn}$ [CHF] | $O\&M_{dhn}$ [CHF/yr] |
|------|------|------|------|------|------|------|------|------|
| 4 @ 40 °C | 135,000 | 3896 | 120,731 | 32,847 | 49,123 | 103,226 | 40,775 | 245 |
| 4 @ 50 °C | 135,000 | 4133 | 140,311 | 33,043 | 55,706 | 103,292 | 51,991 | 319 |
| 4 @ 60 °C | 135,000 | 4262 | 159,930 | 33,239 | 61,788 | 103,353 | 64,860 | 390 |
| 7 @ 40 °C | 234,000 | 6564 | 139,947 | 33,039 | 73,576 | 103,471 | 88,969 | 536 |
| 7 @ 50 °C | 234,000 | 7171 | 167,811 | 33,318 | 85,739 | 103,592 | 114,983 | 696 |
| 7 @ 60 °C | 234,000 | 7643 | 196,622 | 33,606 | 98,368 | 103,719 | 144,599 | 872 |
| 12 @ 40 °C | 399,000 | 7354 | 172,139 | 33,361 | 121,027 | 103,945 | 200,210 | 1208 |
| 12 @ 50 °C | 399,000 | 8081 | 214,565 | 33,785 | 150,903 | 104,244 | 278,127 | 1681 |
| 12 @ 60 °C | 399,000 | 8533 | 256,379 | 34,203 | 181,206 | 104,547 | 361,643 | 2190 |
| 24 @ 40 °C | 795,000 | 15,660 | 217,936 | 33,819 | 200,223 | 104,737 | 417,209 | 2523 |
| 24 @ 50 °C | 795,000 | 17,033 | 278,536 | 34,425 | 262,728 | 105,362 | 606,402 | 3670 |
| 24 @ 60 °C | 795,000 | 17,870 | 345,930 | 35,099 | 333,846 | 106,073 | 834,604 | 5058 |
| 42 @ 40 °C | 1,389,000 | 27,464 | 257,174 | 34,211 | 296,038 | 105,695 | 712,115 | 4310 |
| 42 @ 50 °C | 1,389,000 | 32,031 | 355,584 | 35,196 | 416,608 | 106,901 | 1,116,367 | 6756 |
| 42 @ 60 °C | 1,389,000 | 35,326 | 468,221 | 36,322 | 580,494 | 108,540 | 1,702,572 | 10,316 |

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
