# Peer review of "Techno-Economic Analysis of a Seasonal Thermal Energy Storage System with 3-Dimensional Horizontally Directed Boreholes"

_2673-7264, doi:10.3390/thermo2040030_

Round 1
Reviewer 1 Report
This paper presents a viable technology to store energy. The paper presents comparisons of different borehole configurations and economical analysis. The following points need to be considered.
1- Lines 473 and 613: "Error! Reference source not found" needs to be fixed.
2- Figure 4: the resolution should be increased. The labels are not clear.
3- The discussion section should be supported with comparison tables.
4- The conclusion is too long and needs to be shortened.
5- At the end of the introduction section, the authors should mention the significance (contribution) of this work as bullet points.
Reviewer 2 Report
The paper deals with the analysis of thermal energy soring in soils. The paper is very well organized, it is understandable. I have only few comments:
11. The unit of heat capacities presented in Table: K-1 is missing
22. Eq (1): If the Eq. (1) has any physical meaning, the temperatures should be in Kelvins not °C
33. Several figures are not readable, e.g. Fig. 2, Fig. 5 etc.
44. Errors in reference source are present in the text several times. Please correct it.
55. Is the Eq (4) correct? If yes, then the thermal efficiency will be always more than 1.
